# A non-image-forming visual circuit mediates the innate fear of heights in male mice

Wei Shang[1,4], Shuangyi Xie[1,4], Wenbo Feng[1,4], Zhuangzhuang Li [2], Jingyan Jia[1], Xiaoxiao Cao[1], Yanting Shen[1], Jing Li[1], Haibo Shi [2], Yiran Gu[1], Shi-Jun Weng[3], Longnian Lin[1], Yi-Hsuan Pan [1]✉ & Xiao-Bing Yuan [1]✉

The neural basis of fear of heights remains largely unknown. In this study, we investigated the fear response to heights in male mice and observed characteristic aversive behaviors resembling human height vertigo. We identified visual input as a critical factor in mouse reactions to heights, while peripheral vestibular input was found to be nonessential for fear of heights. Unexpectedly, we found that fear of heights in naïve mice does not rely on image-forming visual processing by the primary visual cortex. Instead, a subset of neurons in the ventral lateral geniculate nucleus (vLGN), which connects to the lateral/ventrolateral periaqueductal gray (l/vlPAG), drives the expression of fear associated with heights. Additionally, we observed that a subcortical visual pathway linking the superior colliculus to the lateral posterior thalamic nucleus inhibits the defensive response to height threats. These findings highlight a rapid fear response to height threats through a subcortical visual and defensive pathway from the vLGN to the l/vlPAG.

Fear of heights is a prevalent and instinctive reaction accompanied by the feeling of anxiety, dizziness, muscle stiffness, and other vegetative symptoms[1]. The physiological reaction to height exposure is considered an innate behavior that serves as a protective mechanism against falling-related injuries[2]. It is believed to affect the entire population to varying degrees. However, an irrational fear of heights can disrupt posture balance, impair motor control, and significantly impact one's quality of life[3]. Approximately 30% of people experience visual height intolerance, a milder form of height-related discomfort. On the more severe end of the spectrum is acrophobia, an intense and debilitating fear of heights that requires clinical intervention. This condition is estimated to affect about 6% of the population, highlighting the need for a better understanding of its underlying mechanisms[3].

Despite its documented presence in ancient literature dating back thousands of years, the fundamental neural mechanisms underlying the fear of heights remain poorly understood. Current theories attribute the physiological fear of heights to posture instability at elevated locations[4–6]. According to these theories, when the distance between an observer and visible stationary contrasts becomes critically large, the effectiveness of visual input in detecting body sway diminishes. This leads to a mismatch between visual and vestibular-proprioceptive information, resulting in heightened posture instability, fear of falling, and increased anxiety[4–8]. This hypothesis suggests that height vertigo is a form of distance vertigo, as both conditions are associated with increased posture instability. However, this hypothesis fails to explain the absence of fear and anxiety when looking at a distance. Therefore, systematic experimental investigations are necessary to determine whether fear of heights arises from direct neural responses to visual stimuli related to height or if it is a consequence of posture instability at height. Moreover, it is important to explore the existence of a specialized brain circuit dedicated to processing height threats.

[1]Key Laboratory of Brain Functional Genomics of Shanghai and Ministry of Education, Institute of Brain Functional Genomics, School of Life Science and the Collaborative Innovation Center for Brain Science, East China Normal University, Shanghai 200062, China. [2]Department of Otolaryngology Head & Neck Surgery, Shanghai Sixth People's Hospital Affiliated to Shanghai Jiao Tong University School of Medicine, Otolaryngology Institute of Shanghai Jiao Tong University, Shanghai 200233, China. [3]State Key Laboratory of Medical Neurobiology and MOE Frontiers Center for Brain Science, Institutes of Brain Science, Fudan University, Shanghai 200032, China. [4]These authors contributed equally: Wei Shang, Shuangyi Xie, Wenbo Feng. ✉e-mail: yxpan@sat.ecnu.edu.cn; xbyuan@brain.ecnu.edu.cn

In this study, we leveraged enriched methodologies for monitoring and manipulating behavior and brain circuit activities in freely behaving mice to unravel the neural mechanisms underlying the physiological response to height threat.

## Results

### Mice display stereotyped aversive reactions to height threat

Adult naive mice placed on an open high platform (OHP, 20 × 20 cm, 30 cm off the ground) exhibit a characteristic 3-phase behavioral pattern during exploration (Fig. 1a, b and Supplementary Movie 1). Initially, they spend approximately 0–9 s exploring the center of the OHP (phase 1, Fig. 1a, b). Subsequently, mice cautiously approach the platform's edges. They extend their heads and forepaws towards the edge while keeping their hind paws in the central region of the platform, resulting in lowering and elongation of the body. Some mice manage to poke their heads out of the platform before withdrawal (out and half-out), while others extend their noses out or simply reach the edge without poking their heads out (nose-in). After a brief edge exploration, animals avoid crossing the edge but either crawl back or turn away from the edge to initiate another round of edge exploration-retraction/turning cycle (phase 2, Fig. 1a, b). After several rounds of edge exploration lasting approximately 0.5–70 s, the mice enter a motionless period lasting around 1–17 s, during which they exhibit visible body trembling (Supplementary Movie 2) and occasional grooming following retraction from the edge (Fig. 1a, b). Once the trembling subsides, they resume the cycle of exploration, retraction/turning, trembling/grooming (phase 3), with variations in the duration of each behavioral condition (Fig. 1a, b). Mice lack vertical exploration (rearing up behaviors) throughout their exploration of the OHP (Fig. 1e). The trembling behavior observed in mice, particularly following edge exploration, differs significantly from the classical freezing behavior seen in fear conditioning (Supplementary Movie 2) and in response to overhead looming threat[9,10]. Unlike freezing, which involves complete body immobilization, trembling is characterized by inhibited limb movements with active head and neck muscles, resulting in visible upper body fluctuations (Supplementary Movie 2).

The stereotyped exploration-retraction-trembling cycle (phase 3) was absent when mice explored same-sized platforms enclosed by either transparent (TWP) or non-transparent (gray, GWP) walls (Fig. 1c, d and Supplementary Movie 3, 4). This suggests that the detection of height threat depends on the head poking over the edge of the high platform. Compared to mice on the OHP, mice on both enclosed platforms exhibited significantly longer locomotion, more frequent self-grooming and vertical exploration, and no visible trembling (Fig. 1f–h). The lowering of the body and the lack of vertical exploration of mice on the OHP align with the human tendency to reduce anxiety by crouching at heights[1]. In rodents, the level of self-grooming is inversely correlated to the aversiveness of tasks at medium to high anxiety levels[11]. Hence, less self-grooming on the OHP than on enclosed platforms also indicates higher anxiety levels on the OHP. Additionally, upon exposure to a height threat, mice exhibited an increase in heart rate, an autonomic response typically associated with heightened anxiety (Supplementary Fig. 1). Male and female mice displayed similar anxiety levels on the OHP (Supplementary Fig. 2).

### Experience-dependent facilitation of fear of heights

When naïve mice were exposed to two trials of 30-min height exposure with a 2-day interval, they exhibited more severe aversive reactions in the second trial. This was evidenced by reduced locomotion, decreased peripheral exploration, less exploration over the platform edge (nose outside), shorter latency for initiating trembling, and longer duration of trembling in the second trial compared to the first trial (Fig. 1i, j and Supplementary Fig. 3). These enhancements of defensive responses were observed regardless of whether the second trial utilized the same OHP in the same environment (Fig. 1i and

Supplementary Fig. 3) or a different-colored OHP in a different environment (Fig. 1j), indicating generalized cross-session facilitation of fear of heights. Interestingly, fear facilitation in the second trial persisted even when the interval between the two trials was as long as 30 days (Supplementary Fig. 3), indicating the presence of a long-lasting memory triggered by a single exposure to height threat. To minimize potential interference of prior height exposure on mouse behavior, all subsequent experiments were conducted using naïve mice.

### Visual input is essential for fear of heights

By comparing the locomotion, the distance explored the peripheral zone, the distance explored over the edge (nose outside), and the trembling duration, we observed a gradual increase in overall fear level with height. The fear level reached saturation when the height exceeded 20 cm (Fig. 2a). The threshold for inducing noticeable body trembling was approximately 10 cm. When the height of the OHP was below 5 cm, most mice jumped or climbed off the OHP within 30 min. However, very few mice were able to descend from the OHP when the height exceeded 10 cm. When naïve mice, without prior exposure to the OHP, were placed on the OHP in the dark, behaviors such as locomotion, peripheral exploration, exploration over the edge, and trembling did not change with height (Fig. 2b), suggesting that fear of heights is visually dependent. In the dark, no mice descended from the OHP, even when it was 5 cm tall (Fig. 2b).

To further investigate the role of visual cues in fear of heights, we compared mouse behavior on the OHP of equal height (30 cm) with or without light during the first two minutes (Fig. 2c–i and Supplementary Movie 5). Mice in the dark explored the edge more often than mice in the light (Fig. 2c, d). Based on how far the animal poked its head over the edge, we divided the edge exploration events into 4 different categories including head-out, half-out, nose-out, and nose-in explorations (Figs. 1b, 2e). Mice in the dark had a significantly higher frequency of head-out and nose-out explorations (Fig. 2c, e, h), but a significantly lower frequency of immediate retraction following head protrusion out (Fig. 2f, h). Moreover, the body trembling following a bout of edge exploration, either immediately following the edge exploration or after the animal has turned back to the platform center, was less frequent in the dark and shorter in total duration than that in the light within the initial 2 min on the OHP (Fig. 2c, g, h).

During exploration, an animal's body lowering with simultaneous elongation indicates cautiousness and anxiety. We noticed that mice in the dark had more frequent body elongation in the platform center and less frequent elongation in the peripheral zone when compared to mice in the light (Fig. 2i). Overall, comparisons of animal behaviors in light and dark supported a vision-dependent fear of heights that is absent in the dark. Furthermore, mice in the dark displayed increased anxiety at the OHP, as indicated by elongated bodies in the center zone (Fig. 2i) and a lack of jumping or climbing off from low platforms (Fig. 2b). However, the anxiety level in the dark did not increase with height, suggesting that it is more likely a fear of uncertainty or posture instability associated with the edge rather than a fear of height.

We also found that mice on an OHP with a transparent base had significantly longer duration of trembling than mice on an OHP with a non-transparent base (Fig. 2j). This finding further supports the positive correlation between the visual height stimulus from below and the fear level at heights. Additionally, monocular vision alone was sufficient to trigger a fear of heights (Fig. 2k).

### Vestibular input is dispensable to fear of heights

Fear of heights was considered to be caused by the conflict between visual and vestibular inputs[5,6]. To investigate the involvement of vestibular input in fear of heights, we first conducted chemical labyrinthectomy by a 2-week treatment with the ototoxic aminoglycoside gentamicin (200 mg kg$^{-1}$, twice a day) via intraperitoneal injection[12]. Gentamicin-treated mice exhibited deficits in the righting reflex

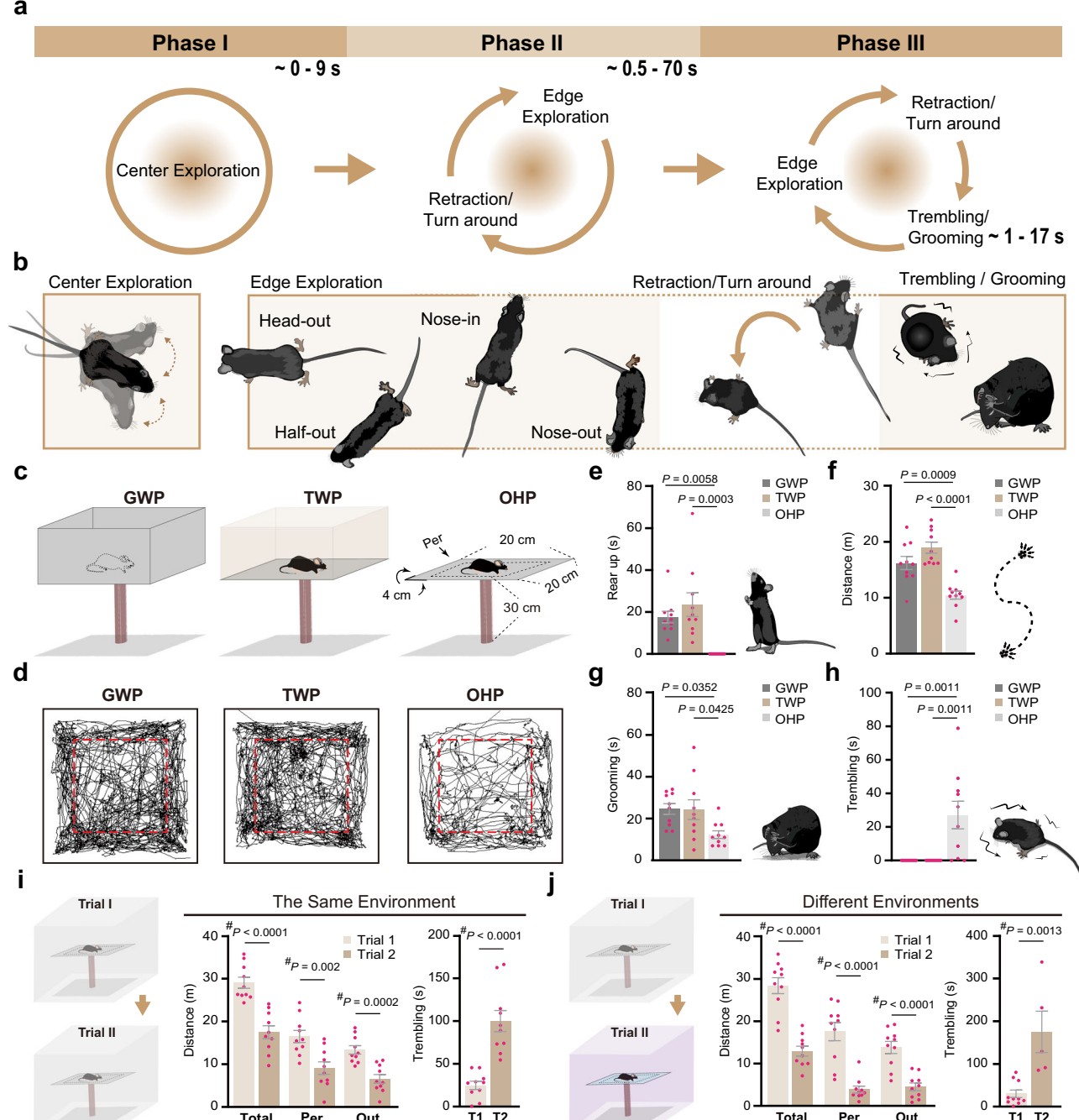

**Fig. 1 | Mice display stereotyped aversive responses to height exposure.**
Different phases (**a**) and typical postures (**b**) of stereotyped behaviors of mice on an open high platform (OHP). **c** Diagrams of enclosed platforms and the OHP used in behavioral tests. The two enclosed platforms, matching the OHP in size and height (20 × 20 × 30 cm), feature either non-transparent (gray) (GWP) or transparent (TWP) walls. The central zone (16 × 16 cm, dashed-line square) and the peripheral zone (Per) are delineated. **d** Representative locomotion traces of mice over 30 min on GWP, TWP, and OHP, with central zone boundaries (red dash lines). Quantitative analysis of mice rearing up (**e**), locomotion (**f**), grooming (**g**), and trembling (**h**) behaviors on three different types of elevated platforms. Cross-session facilitation

of fear of heights. Mice underwent two 30-min exposure trials to the OHP. The color of the OHP and the surrounding environment during the second trial were either identical (**i**) or different (**j**) from that of the first trial. Total locomotion distance (Total), locomotion distance at the peripheral zone (Per), the locomotion distance of the nose tip outside the platform (Out), and the total trembling duration of animals in the two trials of the OHP test are shown in the histograms. Data in (**e**)–(**j**) are presented as mean ± S.E.M. from $n = 10$ mice/group. Pink dots symbolize individual mice. Statistical analyses are One-Way ANOVA with post-hoc $t$-tests in (**e**)–(**h**), and a two-tailed (#) Student's $t$-test in (**i**) and (**j**). Source data are provided as a Source Data file.

(Supplementary Fig. 4a), impaired posture balance during swimming (Supplementary Fig. 4b and Supplementary Movie 6), and a significant reduction in the frequency of moderate and heavy efforts of climbing the pool wall (Supplementary Fig. 4c and Supplementary Movie 7), indicating impaired vestibular function. However, the fear of heights of

gentamicin-treated mice was not affected in the OHP test (Fig. 2l). The open-field test (OFT) and the elevated O-maze test (EOM) were conducted to evaluate the anxiety level of these mice[13,14]. Gentamicin treatment had little effect on the central zone exploration in the OFT (Supplementary Fig. 4d) or the open-arm exploration in the EOM

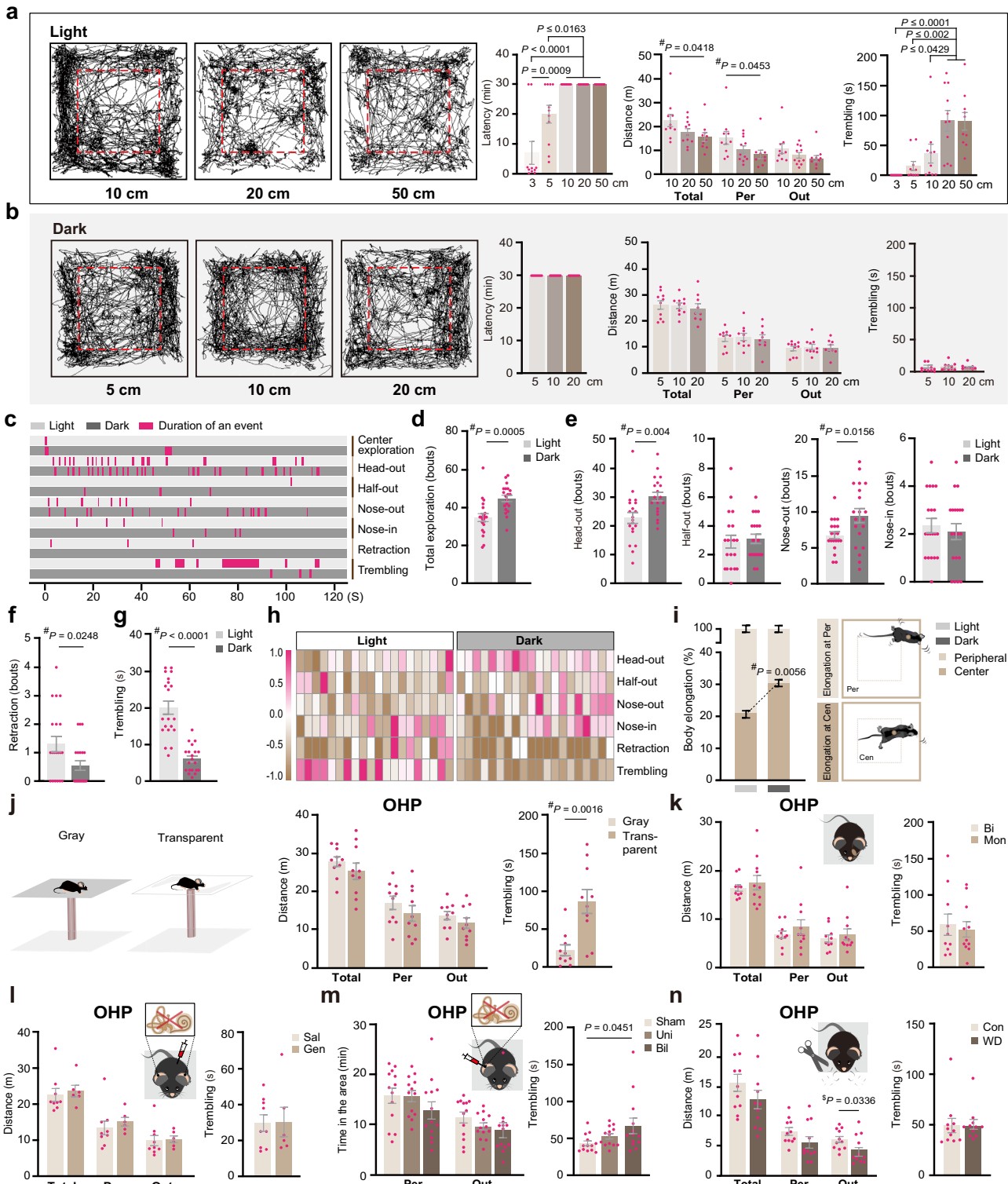

**Fig. 2 | Visual input plays a primary role in the fear of heights. a, b** Locomotion traces (left) and behavior quantification (right) on OHPs (3–50 cm in light; and 5–20 cm in dark). Latency indicates the time taken for the animal to jump or climb off the OHP, measured up to 30 min ($n = 9$, 10, or 11 mice/height). **c** Plots of behavioral events over time for a representative mouse in light and a representative mouse in dark (first 2 min on the OHP). **d** Bouts of peripheral exploration in the initial 2 min under light or dark environments ($n = 20$ mice/environment). **e–g** Behavioral event duration in light and dark ($n = 20$ mice/condition). **h** Behavioral event summary in the first 2 min, with event levels (−1.0 to 1.0, normalized across all animals) shown in pseudo colors per mouse. **i** Events of crawling with body elongation in mice in center vs. peripheral area in light or dark ($n = 20$ mice/environment). **j** Behavior comparison on non-transparent vs. transparent OHP bases ($n = 10$ mice/base). **k** Effects of treatments on the fear of heights ($n = 10$ or 11 mice/treatment). Treatments include monocular visual deprivation, systemic chemical labyrinthectomy (control $n = 9$ and Gen group $n = 6$) (**l**), unilateral/bilateral labyrinthectomy ($n = 13$ mice/group) (**m**), and whisker deprivation ($n = 11$ mice/group) (**n**). Pink dots symbolize individual mice. Data are presented as mean ± S.E.M. with One-Way ANOVA with post-hoc $t$-tests. A one-tailed ($\$$) or two-tailed ($\#$) Student's $t$-test was used. Source data are provided as a Source Data file.

(Supplementary Fig. 4e), only reduced mouse locomotion in the open-field arena (Supplementary Fig. 4d). We further examined the effect of more specific unilateral and bilateral impairment in peripheral vestibular input on the fear of heights by intratympanic application of sodium arsanilate[15] (100 mg ml$^{-1}$). One day after arsanilate application, mice subjected to bilateral treatment exhibited significant deficits in their vestibulo-ocular reflex (VOR, Supplementary Fig. 4g), indicating severe vestibular impairments. Mice with unilateral arsanilate application showed spontaneous nystagmus and a tail suspension circling (Supplementary Movie 8), both typical signs of vestibular imbalance. In both unilateral and bilateral cases, arsanilate application significantly extended the duration of the righting reflex seven days after the treatment (Supplementary Fig. 4h). These sodium arsanilate-treated mice sank and drowned quickly in the swimming test due to severe deficits in vestibular function. We found that both unilateral and bilateral vestibular deficits significantly increased the animal locomotion in the OFT without affecting their center exploration in the OFT (Supplementary Fig. 4i) or their open-arm exploration in the EOM (Supplementary Fig. 4j). In the OHP test, neither unilateral nor bilateral intratympanic treatment with sodium arsanilate significantly affected locomotion or edge exploration of the animal; however, bilateral intratympanic treatment significantly increased the animal trembling on the OHP (Fig. 2m). Moreover, treatment of mice with the vestibular depressant and anti-vertigo agent difenidol (11 mg kg$^{-1}$)[16] failed to alter the fear of heights in the OHP test (Supplementary Fig. 4k). These results do not support an essential role of vestibular input in triggering fear of heights. Finally, shaving whiskers did not have a significant impact on mouse behaviors on the OHP (Fig. 2n), indicating that somatosensory input from whiskers is dispensable for the sensation of height threat in mice.

### The superior colliculus suppresses the fear of heights
To investigate how visual input drives the fear of heights in naïve mice, we conducted a c-Fos-based mapping of neuronal activation in response to height exposure. We focused on the visual pathway and several brain regions known to be involved in innate and conditioned fear responses. Following the mapping of neuronal activation, we manipulated the neuronal activities in these brain regions to examine their effects on animal behavior on the OHP.

The superior colliculus (SC) is a subcortical center for the ocular-motor reflex and innate defensive response to visual threats[17–19]. Glutamatergic projection from the superficial layers of the medial region of SC to the lateral posterior thalamus nucleus (LPTN) has been implicated in freezing response to overhead looming stimuli[18,20,21]. We found that height exposure led to a significant increase in Fos-positive cells in the superficial layers of the medial region of the SC (Fig. 3a, c) and the rostral medial area of the LPTN (LPMR, Fig. 3b, d).

Next, we chemogenetically inhibited neuronal activity in specific brain regions of interest in mice using Clozapine-N-oxide (CNO) treatment (3.5 mg kg$^{-1}$, intraperitoneal injection). The mice were previously injected with the AAV virus encoding the inhibitory construct hM4Di in the targeted brain regions. The effectiveness of neuronal activity inhibition was confirmed using whole-cell patch-clamp recordings in brain slices (Supplementary Fig. 5a, b) and Fos-staining on virus-expressing brain sections (Supplementary Fig. 5c, d). Surprisingly, chemogenetic inhibition of pan-neuronal activities (AAV2/9-hSyn-hM4Di-EGFP) in either the SC (Fig. 3e, f) or its target area LPMR (Fig. 3i, j) significantly enhanced fear of heights in the OHP test, as indicated by reduced locomotion and edge exploration and increased trembling duration. Chemogenetic inhibition of glutamatergic neurons (AAV2/9-Vglut2-hM4Di-EGFP) in the SC produced a similar effect (Fig. 3g, h). Moreover, specific inhibition of the SC-LPMR projecting neurons by systemic application of CNO in mice injected with AAV2/Retro-hSyn-Cre-EGFP in the LPMR and AAV2/9-hSyn-DIO-hM4Di-mCherry in the SC also significantly increased the fear of heights (Fig. 3k, l). A similar increase in fear of heights was observed when we specifically inhibited the axon

terminals of the SC-LPMR projection through intracranial administration of CNO via micro cannulas implanted in the LPMR region (Fig. 3m, n). Chemogenetic inhibition of neurons of the SC also significantly reduced the open-arm exploration of mice in the EOM without affecting the locomotion and central zone exploration of the same group of animals in the OFT (Supplementary Fig. 6a, b). These results suggest that activation of the SC-LPMR circuit by height exposure specifically suppresses fear of heights without affecting the general anxiety level.

### The primary visual cortex is dispensable for fear of heights
For conscious perception of visual input, the retinal signals are transmitted to the primary visual cortex (V1) via the dorsal lateral geniculate nucleus (dLGN). We observed an increase in the number of Fos-positive cells in V1 in response to a 10-min height exposure (Fig. 4a–c). After surgical ablation of V1 (Fig. 4d), mice exhibited impaired context discrimination in the standard fear conditioning test (Fig. 4e), indicating deficits in visual perception of the environment. Unexpectedly, the fear of heights of V1-ablated mice remained unaffected, as shown by a normal level of edge exploration and trembling on the OHP (Fig. 4f, g). Their anxiety levels in the OFT and EOM were also largely unaffected, with only a slight reduction in locomotion in the OFT (Supplementary Fig. 7a, b). In another experiment, mice were injected with the AAV virus carrying the inhibitory vector AAV2/9-hSyn-hM4Di-EGFP specifically in V1. Chemogenetic inhibition of neuronal activities in V1 through systemic injection of CNO had no effect on fear of heights in the OHP test (Fig. 4h, i). These results suggest that the image-forming visual processing by V1 is nonessential for triggering fear of heights in naïve mice.

### Involvement of vLGN in fear of heights
Exposure to heights significantly elevated the number of Fos-positive cells in the ventral lateral geniculate nucleus (vLGN) (Fig. 5a–c), an important part of the caudal prethalamus that receives direct input from intrinsically photosensitive retinal ganglion cells (ipRGCs) and relays non-image-forming visual information[22–24]. Remarkably, approximately 88.4% of these Fos-positive cells in the vLGN were identified as GABAergic neurons (Fig. 5d, e). In contrast, the dLGN, which is responsible for conveying image-forming visual information from the retina to V1, displayed minimal Fos activation (Fig. 5a, c).

To further substantiate the response of GABAergic neurons in the vLGN to height exposure, we performed calcium imaging using fiber photometry (Fig. 5f–h). This experiment involved Vgat-Cre mice injected with a Cre-dependent viral vector coding for the calcium indicator GCaMP6s (AAV2/9-hSyn-DIO-GCaMP6s) in the vLGN. Mice acclimated on a high platform enclosed by a transparent wall were exposed to a height threat by removing the wall. We observed an immediate and significant increase in calcium signal upon height exposure, markedly exceeding the amplitude of calcium fluctuations experienced when mice were exposed to open space at ground level (Fig. 5i–k). These findings indicate that height exposure actively engages GABAergic neurons in the vLGN.

When neuronal activities in the vLGN were chemogenetically inhibited, we noted a significant decrease in fear of heights, evidenced by increased exploratory behavior at the edge and reduced trembling on the OHP (Fig. 5l, m). Chemogenetic inhibition of the vLGN had negligible impact on animal behaviors in the OFT (Supplementary Fig. 8a). Complementing these findings, subsequent experiments on Vgat-Cre mice showed that targeted chemogenetic inhibition of GABAergic neurons in the vLGN similarly resulted in a pronounced decrease in fear and anxiety on the OHP. This outcome was evidenced by a significant increase in the duration of exploratory activities near the edge and a significant decrease in trembling duration on the OHP (Fig. 5n, o). These findings support the hypothesis that the innate fear of heights in naïve mice is predominantly a non-image-forming process, likely mediated through GABAergic neurons of the vLGN.

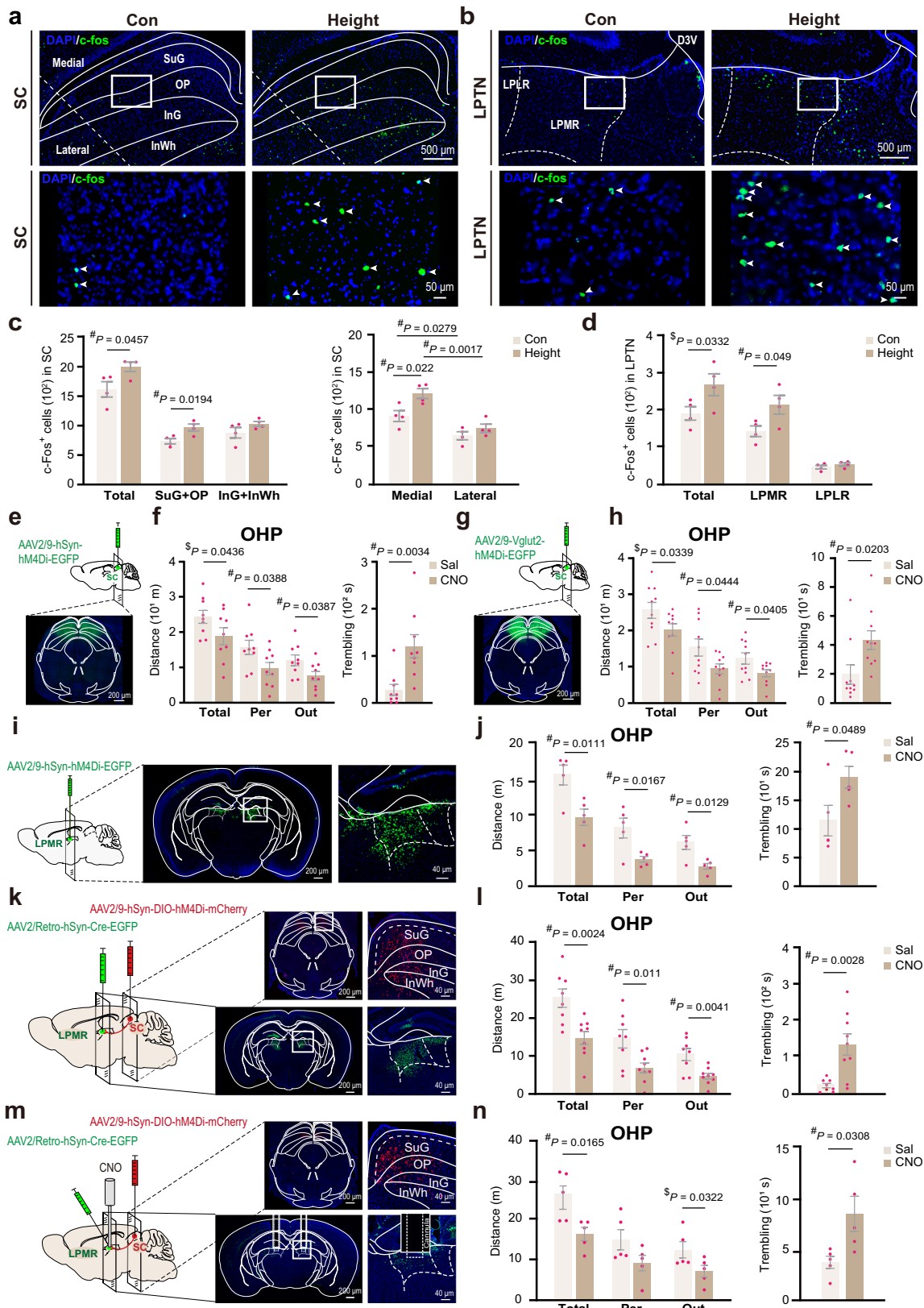

## Glutamatergic neurons in l/vlPAG play essential roles in fear of heights

The vLGN innervates several brain regions associated with the expression of fear and anxiety, including the lateral and ventrolateral regions of periaqueductal gray (PAG)[25]. Immunofluorescent staining revealed that the majority of neurons projecting from the vLGN to the lateral and ventrolateral PAG (l/vlPAG) are GABAergic (Supplementary Fig. 8b), which is consistent with the previous study showing that GABAergic projecting neurons from the vLGN innervate GABAergic interneurons in the l/vlPAG[25].

After height exposure, there was a significant increase in the number of Fos-positive cells in the lateral and ventrolateral PAG (lPAG,

**Fig. 3 | The SC-LPMR circuit suppresses the fear of heights.** Representative images (**a**, **b**) and (**c**, **d**) c-Fos+ cell quantification in the superior colliculus (SC; **a**, **c**) and the lateral posterior thalamic nucleus (LPTN; **b**, **d**) (n = 4 mice/group). LPLR and LPMR refer to the lateral rostral and medial rostral subregions of LPTN, respectively. Chemogenetic inhibition of SC neurons increases the fear of heights in mice (**e**) on the OHP after CNO treatment, as indicated by reduced peripheral exploration and increased trembling duration (n = 9 mice/group) (**f**). **g**, **h** Effect of chemogenetic inhibition of glutamatergic neurons in the SC on fear of heights (n = 10 mice/group). **i**, **j** Effect of chemogenetic inhibition of pan-neuronal activities

at LPMR on the fear of heights (n = 5 mice/group). **k**, **l** Effect of chemogenetic inhibition of SC neurons that project to LPMR on the fear of heights. Virus vectors for Cre-dependent hM4Di (red) were injected into the SC, and AAV2/Retro-hSyn-Cre-EGFP vectors (green) were injected into the LPMR (control n = 8 and CNO group n = 9). **m**, **n** Effect of local inhibition of axon terminals of the SC-LPMR pathway on fear of heights. CNO was applied through a cannula duplex implanted in the LPMR region (n = 5 mice/group). Pink dots indicate individual mice. Data are presented as the mean ± S.E.M. and analyzed using a one-tailed (\$) or two-tailed (#) Student's t-test. Source data are provided as a Source Data file.

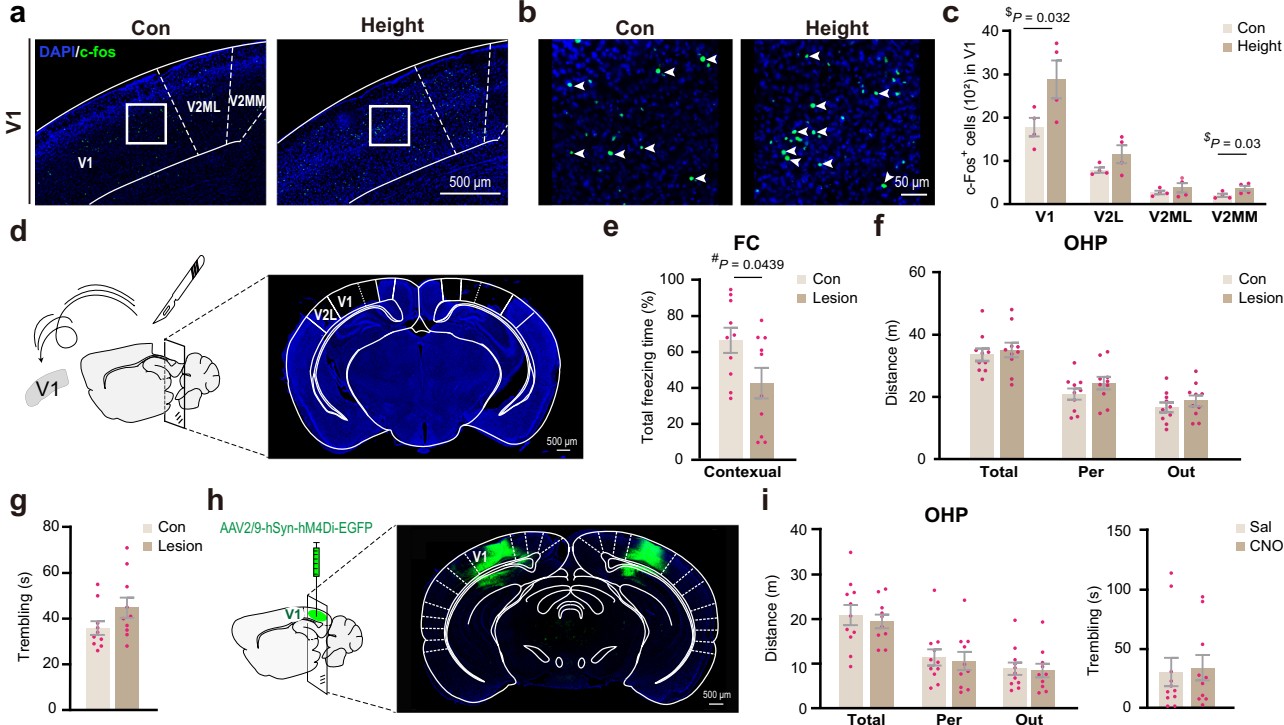

**Fig. 4 | The primary visual cortex is dispensable for fear of heights.**
**a**, **b** Immunofluorescence staining of c-Fos in the visual cortex. A representative section (**a**) and high-magnification images of selected regions (squares) in the visual cortex are shown (**b**). **c** Quantitative analysis of c-Fos+ cells in response to 10-min height exposure in different subregions of the visual cortex (V1, V2L, V2ML, V2MM) in mice (n = 4 mice/group). **d**–**f** Effect of surgical ablation of V1 on fear of

heights. Results of fear conditioning (FC) (**e**) and OHP (**f**, **g**) tests are shown (n = 10 mice/group). **h**, **i** Effects of chemogenetic inhibition of pan-neuronal activities in V1 on the fear of heights (control n = 11 and CNO group n = 10). Pink dots represent individual mice. Data are presented as the mean ± S.E.M. and analyzed via a one-tailed (\$) or two-tailed (#) Student's t-test. Source data are provided as a Source Data file.

vlPAG) (Fig. 6a–c). Calcium imaging via fiber photometry was further performed on the l/vlPAG in mice, specifically targeting glutamatergic neurons expressing the calcium indicator GCaMP6s in this region. Upon height exposure, we noted an immediate rise in calcium signal, significantly surpassing the fluctuation amplitude seen when the same animal encountered open spaces at ground level (Fig. 6d–g). This response suggests that the glutamatergic neurons in the l/vlPAG are actively responsive to height-related threats.

Chemogenetic inhibition of pan-neuronal activities in the l/vlPAG by treating mice with CNO after injecting AAV2/9-hSyn-hM4Di-EGFP in this brain region resulted in a significant disruption of the typical fear responses exhibited by mice on the OHP (Fig. 6h). Most CNO-treated mice did not retract at the edge of the OHP but instead continued to advance and explore underneath the platform until they either fell off the platform or hung on to the edge (Fig. 6i and Supplementary Movie 9). Some mice even jumped directly off the OHP after edge exploration. Additionally, the CNO-treated mice that remained on the OHP had a significantly reduced duration of trembling compared to the saline-treated mice (Fig. 6j). This treatment did not alter animal behavior in the OFT and EOM (Supplementary Fig. 9a, b). Furthermore, chemogenetic inhibition of glutamatergic activities using AAV2/9-Vglut2- hM4Di-EGFP

in the l/vlPAG also caused most mice to get off the OHP (Fig. 6k, l). Additional experiments were carried out using Vglut2-Cre mice to examine the effects of bidirectional manipulation of glutamatergic activity in the l/vlPAG (Fig. 6m). In these mice, chemogenetic inhibition of glutamatergic neurons led to most of them leaving the OHP, while chemogenetic activation resulted in a freezing-like behavior on the OHP (Fig. 6n; Supplementary Movie 10). These findings indicate the crucial role of the l/vlPAG and its glutamatergic activity in the fear expression associated with heights.

To clarify whether visual height stimulus activates the PAG through the vLGN to induce fear of heights, we conducted chemogenetic inhibition of vLGN neurons that project to the l/vlPAG. This was achieved by systemic CNO treatment of mice after injecting them with the Cre-dependent inhibitory construct (AAV2/9-hSyn-DIO-hM4Di-mCherry) in the vLGN and AAV2/Retro-hSyn-Cre-EGFP in the l/vlPAG (Supplementary Fig. 10a). This treatment significantly alleviated fear of heights, as evidenced by a significant increase in edge exploration and a decrease in trembling duration compared to mice treated with saline (Supplementary Fig. 10b, c). This treatment did not affect animal behavior in the OFT and EOM (Supplementary Fig. 10d, e), indicating specificity of the effects to fear of heights. Additionally, the local

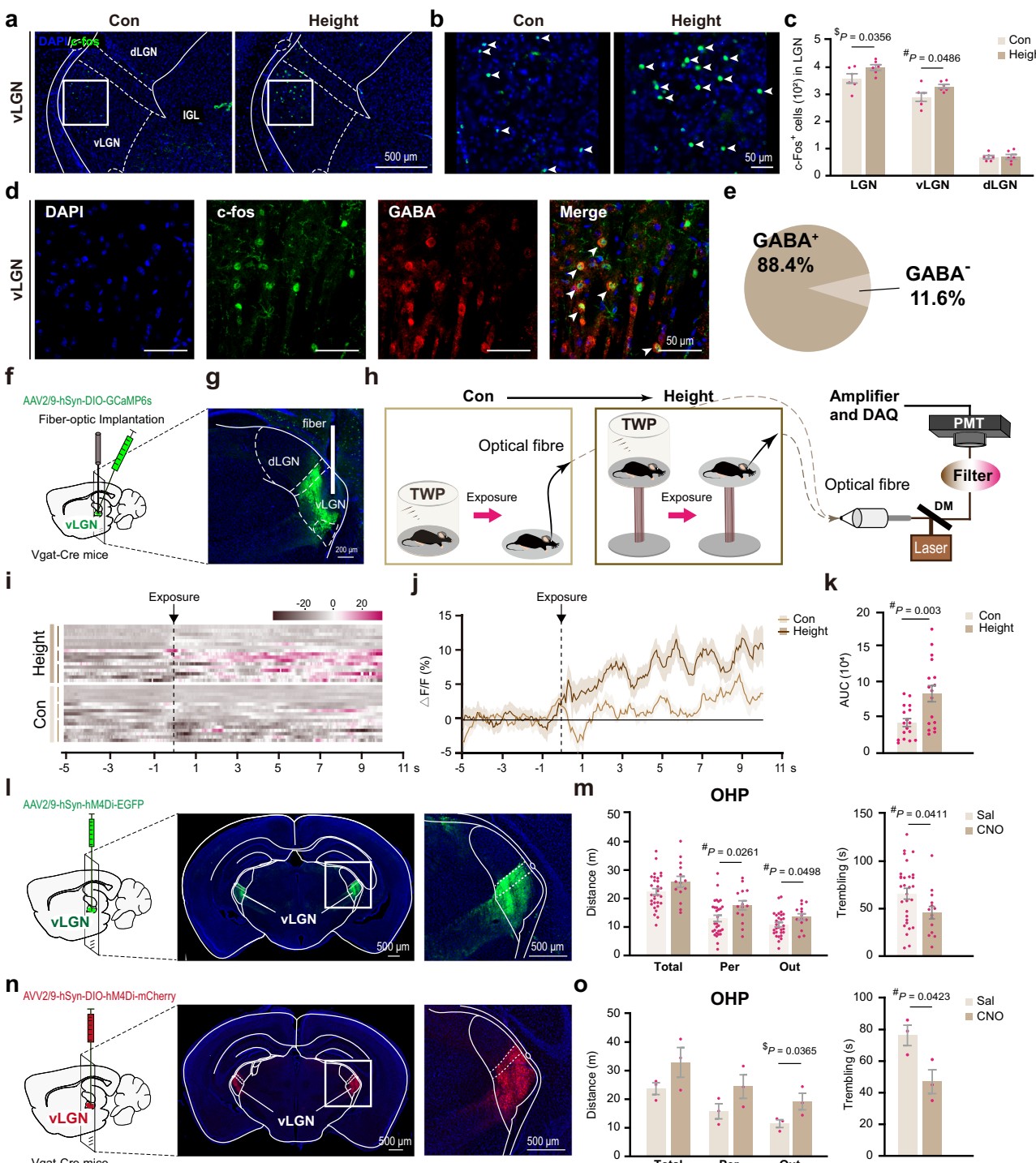

**Fig. 5 | vLGN is essential for fear of heights. a**, **b** c-Fos immunofluorescence in sections of the LGN from control and height-exposed mice. **c** Counts of c-Fos[+] cells in subregions of the LGN (total, vLGN, and dLGN) (*n* = 6 mice/group). **d** Immunofluorescent co-staining of c-Fos (green) and GABA (red) in the vLGN of mice exposed to heights. Arrowheads indicate double-positive neurons. **e** The pie chart illustrates the proportion of double-positive cells among total c-Fos[+] cells from *n* = 2 mice. **f**–**k** Calcium imaging of GABAergic neurons in vLGN. Cre-dependent GCaMP6s virus vectors were injected into the vLGN of Vgat-Cre mice, and optic fibers were implanted (**f**, **g**). Schematic of fiber optic recording. The same mouse was first exposed to open space at ground level as a control condition, followed by height exposure (**h**). Heatmaps (**i**) and time-based Ca²⁺ fluctuations (**j**) during control and height exposures are presented. Each short bar to the left of (**i**) indicates one mouse (*n* = 3 mice). **k** Comparison of the Ca²⁺ rise amplitude, based on the area under curve (AUC), calculated from (**j**). **l**, **m** Effects of pan-neuronal inhibition in vLGN on height fears (control *n* = 30 and CNO group *n* = 14). **n**, **o** Effects of GABAergic neuron inhibition in vLGN on fear of heights (*n* = 3 mice/group). Schematics of the virus vector injections are shown in (**l**) and (**n**), with OHP test results in (**m**) and (**o**). Pink dots symbolize individual mice. Data are presented as the mean ± S.E.M. with a one-tailed (\$) or two-tailed (#) Student's *t*-test. Source data are provided as a Source Data file.

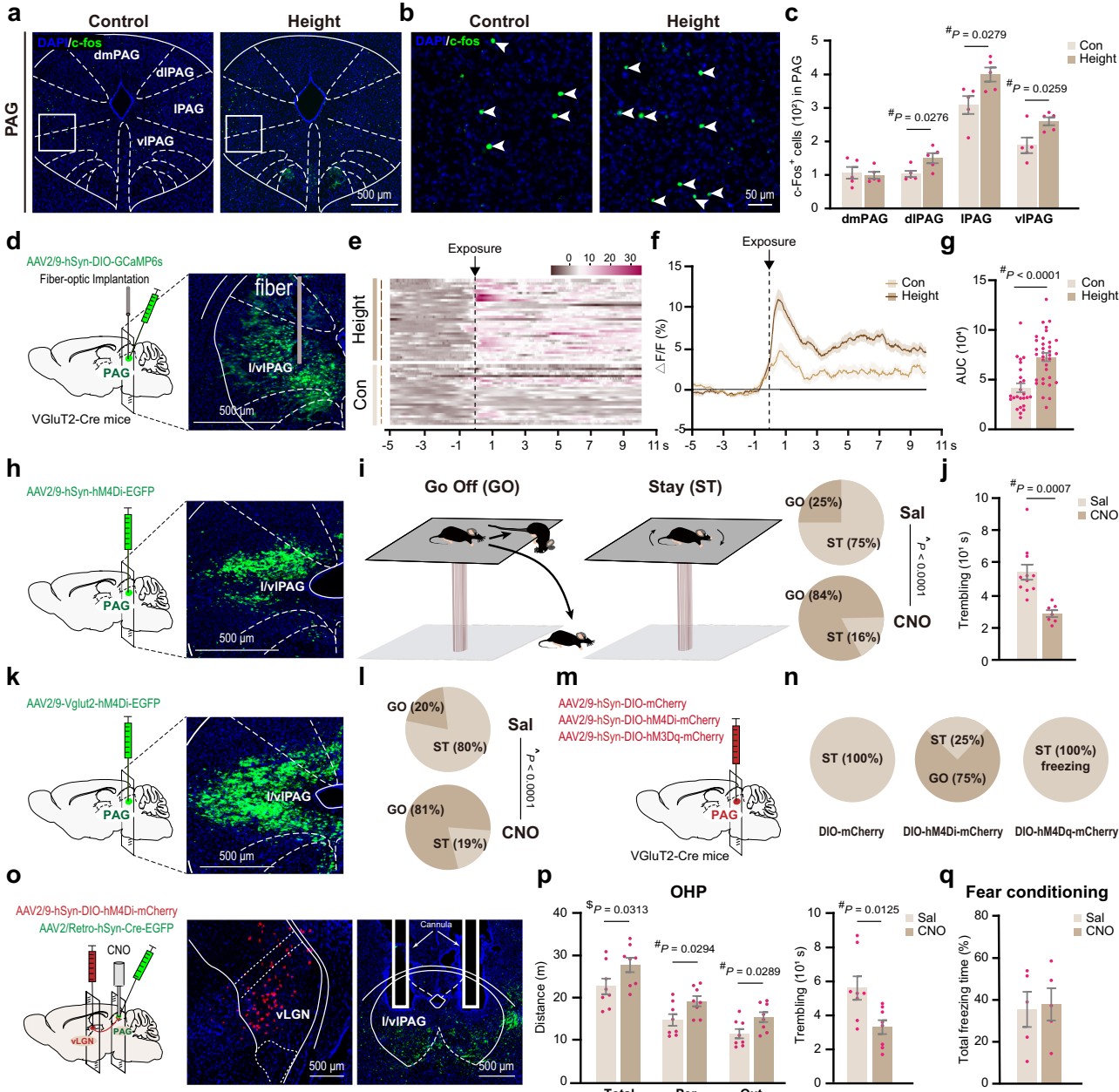

**Fig. 6 | The vLGN-PAG circuit mediates the fear of heights. a–c** c-Fos+ cells in PAG subregions (dmPAG, dlPAG, lPAG, vlPAG) after height exposure (*n* = 5 mice/group). Calcium imaging of glutamatergic neurons in l/vlPAG. Cre-dependent GCaMP6s virus vectors were injected into the l/vlPAG of Vglut2-Cre mice, and optic fibers were implanted (**d**). Heatmaps (**e**) and time-based Ca²⁺ fluctuations (**f**) during control and height exposures are presented. Each short bar to the left of (**e**) indicates one mouse (*n* = 6 mice). **g** Comparison of the Ca²⁺ elevation amplitude based on the area under the curve (AUC) calculated from (**f**). Pan-neuronal (**h**–**j**) and glutamatergic (**k**, **l**) inhibition effects in l/vlPAG on height fear, with Go and ST (**i**–**l**) indicating OHP descent/stay outcomes at the end of the 30-min test (control *n* = 12,

11, and 10, and CNO group *n* = 14, 7, and 11 in **i**, **j**, and **l**, respectively). **m**, **n** Effects of glutamatergic neuron modulation in l/vlPAG on the fear of heights, shown with pie charts. **o**–**q** Local inhibition of vLGN-PAG axon terminals, with virus injections in vLGN and l/vlPAG. CNO was applied through cannula duplets implanted in the l/vlPAG region (**o**). Results of OHP tests and contextual fear conditioning tests are shown in (**p**) and (**q**) (*n* = 8 mice/group in **p** and 5 mice/group in **q**). Pink dots symbolize individual mice. Data are presented as the mean ± S.E.M. with a one-tailed ($) or two-tailed (#) Student's *t*-test. The chi-square test (^) is used in (**i**) and (**l**). Source data are provided as a Source Data file.

infusion of CNO into the l/vlPAG region to specifically inhibit the presynaptic terminal of the vLGN-PAG pathway also significantly reduced the fear of heights, while not impacting responses in the classical fear conditioning test (Fig. 6o–q). This finding supports the notion that this specific pathway is involved in the expression of fear of heights, rather than playing a general role in the response to aversive stimuli.

Contrasting with the pivotal role of the PAG in fear of heights, we observed no significant increase in the Fos signal in the lateral

amygdala (LA) and basal amygdala (BA) following height exposure (Supplementary Fig. 11a−c). These regions are known centers for fear and anxiety responses to visual threats. Although a small population of neurons in the basolateral amygdala (BLA) has been noted to respond to height exposure[26], chemogenetic inhibition of BLA activities resulted in elevated fear of heights. This was evidenced by decreased exploration of the peripheral zone and outside the OHP, along with prolonged trembling (Supplementary Fig. 11d). Conversely, inhibiting the central nucleus of the amygdala (CeA), a major output region of the

amygdala, did not affect fear of heights (Supplementary Fig. 11e). Thus, the PAG, instead of the BLA, plays a significant role in the expression of fear of heights.

## Discussion

The aversive responses of naïve mice to height threats, such as withdrawal, trembling, low body position, absence of vertical exploration, reduced locomotion, increased heart rate, and experience-dependent regulation, bear a striking resemblance to human responses in similar situations. These behavioral parallels across species strongly imply that the fundamental mechanisms of fear of heights are phylogenetically conserved. Thus, mice serve as an excellent animal model for dissecting the neural mechanisms of height fear and for developing translational approaches to address acrophobia. Notably, laboratory mice have no prior experience of falling trauma or distressing memories associated with heights. Their fear of heights is likely an innate response to height stimuli, potentially mediated by a hardwired brain circuit that facilitates adaptation and survival. Consistent with this notion, retrospective self-reports and longitudinal studies in humans have shown that the acquisition of height fear is primarily a non-associative processes[27]. Moreover, the observed cross-session facilitation of fear of heights in mice suggests an experience-dependent modulation of this fear response, potentially involving plastic changes of the relevant brain circuit. This modulation may depend on the cognitive perception of height threats upon repeated exposure. Hence, fear of heights in a natural environment may include both an innate response and an experience-dependent component. The cross-session facilitation, instead of habituation, of fear of heights, underscores the importance of carefully designing exposure protocols in exposure therapy for acrophobia to avoid fear reinforcement from repeated exposure. It aligns with the established principles of exposure therapy, which involves gradual exposure in a safely controlled environment, starting with less anxiety-inducing heights, instead of immediate exposure to challenging heights[28–30].

We discovered that mouse behavioral response to height exposure saturates at approximately 20 cm, mirroring the saturation observed in human fear of heights[5,31]. This similarity not only underscores the physiological relevance of our experimental model but also sheds light on how emotional responses to stimuli like heights may have a saturation point. Generally, animal's responses to stimuli exhibit a non-linear pattern with a saturation point, possibly a protective survival mechanism. This saturation could be influenced by various factors, including the sensory input. The visual input from the lower field, which is crucial in eliciting fear at heights, may have a saturation point beyond a certain altitude. At the neuronal level, it is conceivable that the non-linear properties of neuronal membranes and their maximum firing rates might play a role in this saturation. Moreover, at the circuitry level, the interplay between fear-enhancing and fear-reducing neural pathways, such as the vLGN-PAG and SC-LPMR pathways we described, may also contribute to regulating this saturation, potentially balancing the emotional response to prevent excessive fear. Overall, these suggest that the saturation in fear of heights could stem from a multifaceted neural interplay, although direct evidence for these mechanisms remains to be established.

Fear of heights was attributed to posture instability resulting from a lack of visual parallax cues and a mismatch between visual information and vestibular and proprioceptive inputs[3–6]. Consistent with this notion, vestibular input influences the distance judgment at height[32,33]. Vestibular exercise has been shown to improve dizziness and body sway in a group of acrophobic patients[34]. Individuals with a complete loss of labyrinthine function are seemingly immune to visually-induced motion sickness[35]. However, recent epidemiological investigations have challenged this notion. Individuals with bilateral vestibulopathy, despite experiencing frequent falling due to impaired balance control, do not exhibit higher susceptibility to acrophobia[36] or increased anxiety levels compared to the general population[37]. These findings suggest that peripheral vestibular input is not essential for the fear of heights, although intact vestibular function may modulate the emotional response to height stimuli through mechanisms that are not yet fully understood. In our study, we made an intriguing observation in mice, where bilateral vestibular deprivation slightly but significantly intensified the fear of heights instead of alleviating it, while unilateral deprivation had no such effect. This suggests that vestibular input, in general, does not account for the fear of heights. Instead, our findings support the notion of a direct visual mechanism underlying the fear of heights, rather than it being a consequence of body imbalance at elevated locations. Thus, the brain mechanisms involved in height vertigo appear to differ primarily from those associated with motion sickness.

Mice displayed increased anxiety when exploring the edge of the OHP, even in darkness. However, unlike in the well-lit environment, anxiety levels did not increase with height in the dark. Hence, edge-associated anxiety includes both a fear of heights which depends on visual input and a vision-independent element likely linked to posture instability at the edge, potentially involving vestibular and proprioceptive inputs. It is important to note that this vision-independent anxiety should not be considered a fear of heights but may intensify the fear and anxiety at heights.

A subcortical pathway through the SC and the LPTN (pulvinar) to the amygdala is believed to mediate the non-conscious processing of emotional visual stimuli in primates[38,39]. In rodents, the SC-LPTN circuit plays a pivotal role in the defensive response to overhead visual threats, including the looming stimuli[17,18,20]. Interestingly, we found Fos activation in a subgroup of neurons within both SC and LPMR following height exposure. To our surprise, chemogenetic inhibition of the SC-LPMR circuit enhanced, rather than blocked, the fear of heights. Moreover, the chemogenetic inhibition of the BLA, a downstream target of the LPTN, resulted in behavioral effects like those observed with the independent inhibition of either the SC or the LPMR, suggesting that the SC may modulate the defensive response to height threats through LPMR and BLA. Considering the LPTN's broad cortical and subcortical projections beyond its connection to the BLA[40], the SC-LPMR pathway might also engage other brain areas to fine-tune the behavioral response to height threats. Animals at heights require more precise motor control to mitigate the risk of falling and navigate away from elevated areas. Their motor response to height threats is distinct from the 'flee' or 'freeze' reactions typically observed in response to looming threats. This necessitates the engagement of additional cortical and subcortical brain regions for both conscious and subconscious regulation of motor and posture control, aiding in survival and adaptation. The widespread connectivity of the LP suggests its potential role as an integrative center for modulating responds to height threats.

It is worth noting that the looming stimulus represents a visual threat from above, whereas the height stimulus is a visual threat from below. We observed that SC neurons projecting to the LPMR predominantly reside in the medial region of the SC, as shown in Fig. 3m. Notably, approximately 59.32% of these neurons are positive for parvalbumin (Supplementary Fig. 6c). Moreover, height exposure led to a significant increase in c-Fos$^+$ cells within the superficial layers of the medial SC, as depicted in Fig. 3a–d. The localization and characteristics of these height-responsive cells closely mirror those SC neurons mediating the looming response, which predominantly originate from the superficial layers of the medial SC, project to the LPTN, and are largely parvalbumin-positive[17,20]. This suggests the potential existence of a shared group of neurons in the medial SC that respond to both types of stimuli. However, it appears that specific neuronal ensembles within these pathways may be selectively activated by different stimuli, resulting in divergent behavioral outcomes. Further investigation is required to fully understand the differences between the neuronal ensembles responsive to height stimuli and those responsive to looming stimuli within the SC.

We made an intriguing discovery that neither V1 nor SC is necessary for the physiological fear of heights in naïve mice. This finding suggests the existence of a subconscious and non-image-forming visual pathway for the onset of fear of heights. It appears that exposure to heights acts as an imminent threat, prompting an immediate defensive response even without visual awareness. Supporting this idea, previous research has demonstrated that the inactivation of V1 only mildly impacts mice's response to looming stimuli[41], a behavior indicative of an innate defense mechanism. However, it is important to note that our findings do not exclude the potential contribution of conscious perception of height threat to the experience-dependent fear of heights. The underlying brain mechanisms involved in the experience-dependent regulation of fear of heights require further investigation.

According to current understanding, distal threats typically elicit activity in the prefrontal cortices, which may reflect the cognitive planning of avoidance strategies. As the threat becomes closer or more proximal, midbrain structures such as the PAG dominate the defensive responses[42]. In line with this concept, our investigation revealed a specific circuitry connecting the vLGN to the l/vlPAG, which mediates the fear of heights. Notably, we observed that the majority of vLGN neurons projecting to l/vlPAG are GABAergic. A previous study has demonstrated that the vLGN GABAergic neurons innervate GABAergic neurons in l/vlPAG, leading to the activation of PAG glutamatergic neurons during pain regulation triggered by light stimulation[25]. It is plausible that a similar disinhibition mechanism occurs when vLGN is activated upon height exposure, ultimately activating l/vlPAG.

Recent research has highlighted the caudal prethalamus as a pivotal region for integrating sensory-motor information and regulating various instinctive behaviors. This region, encompassing the vLGN, intergeniculate leaflet (IGL), the thalamic reticular nucleus (TRN), and the zona incerta (ZI), receives direct inputs from the sensory periphery and maintains extensive interconnections with numerous areas across the forebrain, midbrain, and hindbrain. Such a unique connection pattern positions the prethalamus nuclei as central hubs in the orchestration of a wide range of instinctive behaviors, including pain, sleep, feeding, hunting, exploration, and defensive responses[43]. As a critical component of the caudal prethalamus, the vLGN is known for its bidirectional control of escape responses to visual threats, mediated by distinct populations of neurons and their specific downstream targets[44,45]. Previous studies have demonstrated that inhibition of GABAergic neurons in the vLGN significantly increases escape behaviors in response to looming stimuli and elevates anxiety levels in OFT[44,45]. We found that both pan-neuronal and GABAergic-specific inhibition of the vLGN markedly diminishes defensive responses to height threats. This adds further evidence to the crucial role of the vLGN in modulating instinctive behaviors, suggesting a predominant involvement of its GABAergic neurons in regulating fear of heights. Our findings revealed that pan-neuronal inhibition of the vLGN did not notably impact behaviors in the OFT, contrasting with previous studies that reported increased anxiety in mice during the OFT following the inhibition of GABAergic neurons in the vLGN[44,45]. This could indicate that the vLGN's glutamatergic neurons might counterbalance the effects of GABAergic neurons in modulating anxiety levels in open-field environments, a hypothesis that merits further investigation.

Neurons in vLGN exhibit a large receptive field and a preference for bright over dark stimuli[22,23], unlike SC neurons, which are activated by shadows[23]. The dendritic arbors of vLGN neurons often display an asymmetric distribution, positioned to collect input from large yet discrete territories[23]. Notably, when exposed to heights, a broad and bright stimulus is generated in the lower visual field, corresponding to the upper retina. Intriguingly, a specific population of GABAergic ipRGCs is enriched in the dorsal temporal quadrant of the retina, and

their axons project to the vLGN[24]. Whether these GABAergic ipRGCs contribute to the coding of visual height stimuli remains to be investigated.

Among the several treatments that effectively alleviated fear of the OHP, chemogenetic inhibition of l/vlPAG displayed the most striking effect. It is conceivable that, in addition to the vLGN-PAG pathway, other yet unidentified brain circuits converge on l/vlPAG to generate the fear of heights. Previous studies have demonstrated that neurons projecting from the SC to the dorsal PAG are involved in the escaping behavior of mice in response to looming stimuli[41]. Our findings, however, suggest that SC activation in response to height stimuli may suppress, rather than provoke, the defensive response to height threats. Given the well-known role of the PAG in mediating fear and anxiety[46], it appears that the SC might not be the direct upstream driver of PAG during the fear response to heights.

Our research uncovered two distinct pathways with contrasting effects in response to height exposure: the vLGN to l/vlPAG pathway, which enhances the expression of fear of heights, and the SC to LPMR pathway, which attenuates it. The extensive interconnections among these brain regions, such as the SC's projections to the PAG and the vLGN's projections to both the SC and LPTN[43], imply potential complex interactions between these pathways. These interactions likely serve to fine-tune the animal's behavioral response to height stimuli in varying environmental contexts. Of particular interest are the vLGN's projections to the SC and LPTN. With most of the vLGN's output neurons being GABAergic, it's plausible that the vLGN-SC and vLGN-LP projections provide feedforward inhibition to the SC-LP pathway. This inhibition could potentially dampen the SC-LP pathway's suppression of height fear, thereby amplifying the fear response to height threat. However, further investigation is necessary to confirm these hypothesized interactions and to fully understand their impact on fear responses induced by heights.

Finally, significant increases in c-Fos-positive cells were also observed in several brain areas associated with mood and behavioral regulation, including the ventral tegmental area (VTA), ventromedial hypothalamus (VMH), and anterior cingulate cortex (ACC) (Supplementary Fig. 12). These findings suggest a potential role for these regions in the arousal and cognitive processing of height threats, contributing to the animals' complex behavioral responses.

## Methods

### Ethics approval
Mouse care and experiments were performed according to the guidelines for the Care and Use of Laboratory Animals of the National Institutes of Health. All animal procedures were approved by the Animal Care and Use Committee of East China Normal University (m20210411).

### Animals
Two-month-old C57BL/6J mice were purchased from Shanghai Jihui Laboratory Animal Care Co., Ltd. Vglut2-Cre mice (#: 028863) and Vgat-Cre mice (#: 028862), also two months old, were obtained from the Jackson Laboratory. All mice were bred within an SPF (Specific pathogen-free) barrier system and were housed in groups of five per cage with free access to food and water, and subjected to a 12-h light/dark cycle. Except for the experiments in Supplementary Fig. 2, male mice were used. These mice underwent an acclimation period of at least one week before being used in any experiments.

### Behavioral tests
Most behavioral assays were conducted on 2–5-month-old male mice. All behavioral tests were conducted during the daytime. Clean the surface of the testing apparatus with 10% ethanol between trials. Wait at least 5 min before the next test to allow ethanol evaporation and odor dissipation. The general order of behavioral tests was the

high-platform test, open-field test, and the elevated O-maze, followed by additional behavioral tests as needed.

**Open-field test.** The open-field test (OFT) is a behavioral paradigm based on a mouse's natural aversion to open areas. The level of anxiety is inferred from the animal's propensity to explore the central area of a square open-field arena, with lesser exploration indicating higher anxiety. Mice were tested using a 4-stop open-field system (Topscan, CleverSys Inc.). They were gently placed in the middle of the arena (40 × 40 cm) and allowed to freely explore for 30 min. Animal locomotion was video-tracked using an automated tracking system (TopScan). The geometric center of the body was used to track the motion of the animals. The software also recorded the duration of movement, the total distance covered during movement, and the time spent in the center of the arena (25 × 25 cm).

**Elevated O-maze test.** The standard elevated O-maze apparatus consists of two open arms and two closed arms of equal length, spaced apart and forming a 5 cm wide circular corridor. The maze is 48 cm in diameter and 50 cm above the ground. Anxious animals typically exhibit a preference for the enclosed, safer sections over the exposed, open areas. The test mouse was gently placed at one end of a closed arm, with its head facing the closed arm, and was allowed to freely explore for 30 min. Animal behaviors were recorded and tracked using the TopScan system. The time spent by the animal inside the two open arms and the frequency of open-arm entry were recorded by the software.

**Open high-platform test.** Naïve mice were gently placed in the middle of the high platform (20 × 20 cm, 30 cm off the ground) made of Plexiglas. A 12 × 12 cm area in the middle of the platform was defined as the central zone, and the rest was defined as the peripheral zone. Animal behaviors were recorded and tracked using the TopScan system. The geometric body center was tracked to indicate animal locomotion inside the platform, and the nose tip was tracked to indicate exploration over the platform edge. The software system automatically recorded and provided data on the distance and total duration of mouse locomotion in each designated region. The self-grooming, rearing up (not including the wall-supported rear-up events), and trembling of mice were manually analyzed based on the video.

**Height exposure for Fos-staining.** Mice were acclimated for 4 h inside a 34 × 12 cm platform enclosed by non-transparent walls with free access to food and water. Subsequently, the surrounding walls were quickly removed to expose the animal to height threat or replaced with transparent walls as a control treatment. After 10 min, the platform was enclosed with non-transparent walls again, and the animals were perfused for immunofluorescence staining after staying there for 1.5 h.

**Righting reflex test.** Mice were put into a transparent Plexiglas tube which is 15 cm long and 3 cm in diameter. They could roll over but could not turn around or rear up (Fig. S4A). Gently turn the tube until the animal is in the supine position. Measure the time for the animal to revert to the upright position. Each mouse was tested for 3–4 trials to obtain the average righting time.

**Swim test.** A 40 × 20 × 12 cm (L × W × D) swimming pool was used for the swimming test. The water temperature was maintained at about 28 °C. The mouse was gently placed in the middle of the pool. Behaviors of the animals for 1 min were video-recorded with the Topscan System, and each animal was tested for 3 trials. An oblique swimming style was quantified as the average swimming time with one side of the body leaning. For quantitative analysis of the wall-climbing efforts of the mice in the swimming pool, a brief touch of the wall by the nose or one forepaw was defined as a light climbing event; several attempts to climb the wall with two forepaws was defined as a medium climbing event, and continued efforts to climb the wall with two forepaws were defined as a heavy climbing event. Light, moderate, and heavy climbing events of each mouse during the 1-min test were counted to calculate their proportions. Mice with severe vestibular dysfunction who failed to swim with their heads outside the water were taken out of the pool to avoid drowning.

**Contextual fear conditioning test.** The contextual fear conditioning test was performed using a method similar to a previous report[10] with minor modifications. Mice were handled 10 min per day for 3 days before the tests. Fear conditioning was conducted in a chamber (25 × 25 × 25 cm; LE116, Panlab, Spain) with a stainless-steel grid floor. Each mouse was placed in the chamber and was allowed to explore freely for 5 min per day for 3 days to acclimate to the chamber. Twenty-four hours after the acclimation, a 30-s sound of 85 dB was presented through a speaker set on top of the conditioning chamber, which served as the conditioning stimulus (CS). During the last 2 s of the CS presentation, mice received a footshock (0.75 mA, 2 s), which served as an unconditioned stimulus (US). Animals were returned to their home cages 1.5 min after the CS-US pairing. Twenty-four hours after the conditioning, contextual fear memory was tested for 4 min in the same chamber. The percentage of time of freezing of the animal was provided by the software (Packwin, panlab, Spain).

## Heatmap construction
The heatmap diagram was constructed to visualize the frequency of different behaviors of mice on the open high platform under light or dark conditions. For each behavior/posture, the total counts or duration of each mouse for all 20 mice tested (10 mice in the light and 10 in the dark) were ranked and normalized to the −1 to 1 range, where 1 represents the mouse with the highest rank of the behavior and is colored in red, and −1 indicates the mouse with the lowest rank of the behavior and is colored in brown. The pseudo-color heatmap was generated using R software.

## Visual cortex lesion
The animals were anesthetized with sodium pentobarbital (70–80 mg kg⁻¹) and transferred to a stereotaxic frame. The skull covering the visual cortex was thinned and removed using a 0.5 mm dental drill (78040, RWD, China). Sterile PBS was applied to hydrate the exposed brain area. Under a microscope and illuminator, a cut of 1 mm deep was performed around the outline of the primary visual cortex (V1) using a microsurgical blade, and the V1 cortical tissue was removed. After being washed with PBS to remove blood, the lesioned area was covered with an absorbable gelatin sponge (Jiangxi Xiangen Medical Technology Development Co., Ltd), and the scalp was sutured using surgical sutures. For the sham control, the scalp was sutured after the skull was removed without the V1 lesion.

## Whisker shaving
Mouse anesthesia was induced with isoflurane. Whiskers were cut from the base using a 10 cm surgery scissor. The mice were left undisturbed for 3 days before the behavioral experiment.

## Monocular deprivation
After anesthesia with sodium pentobarbital (70–80 mg kg⁻¹), mice were placed on top of a warming pad, and cleaned with alcohol pads. Under a microscope and illuminator, the eyelids were sutured using surgical sutures. Two to three stitches were made for each eye, the suture was tightened, two knots were tied, and extra sutures were cut as short as possible with ophthalmic scissors. A thin layer of quick-dry glue was applied to the eyelid sutures to avoid the loosening of the suture. If bleeding occurs, stop it with cotton before proceeding.

The mice were left undisturbed for 3 days before the behavioral experiment.

## Pharmacological inhibition of vestibular function

Mice received an IP injection of the vestibular suppressant and anti-vertigo agent, difenidol (Shanghai yuanye Bio-Technology Co., Ltd, China), at 11 mg kg$^{-1}$. Wait for 20 min before conducting the behavioral tests.

## Chemical labyrinthectomy

**By systematic application of gentamicin.** Mice received a daily intraperitoneal (IP) injection of 200 mg kg$^{-1}$ of gentamicin (No. 56688, Rongjiarun Origin, China) twice a day for 2 weeks. They were left undisturbed for one additional week for adaptation before being tested. The vestibular deficits caused by gentamicin treatment were validated by the righting reflex test and swim test.

**By intratympanic application of sodium arsanilate.** A 1 ml syringe with a 34-gauge needle was used to penetrate the tympanic membrane to inject 100 mg ml$^{-1}$ sodium arsanilate (A9528 Sigma-Aldrich, 40 μl, prepared in 0.3 M NaHCO$_3$) into the middle ear. Mice were left undisturbed for one week before behavioral tests. The vestibular deficits of treated mice were validated by the righting reflex test.

## Vestibular function assessment

The vestibulo-ocular reflex (VOR) was assessed using a VOR testing system for mouse (GAT-MVOR943, Giant Technology Co., Ltd, Shenzhen, China) two days post intratympanic administration of sodium arsanilate. To restrict body movement, mice were secured in a custom-built chamber on a motorized rotational platform. Eye movements were captured at 60 frames per second using side cameras under infrared illumination, concurrent with the 1.0 Hz horizontal rotation of the platform (−40° to 40°). To enhance pupil tracking, pilocarpine nitrate eye drops (0.02 g ml$^{-1}$) were applied for pupil constriction. The analysis of the pupil movements and the VOR gain (the ratio of eye movement to head movement) were determined using a custom MATLAB2019 script. To record spontaneous nystagmus, the same setup was utilized without rotation of the platform.

## Microinjection

After anesthesia with sodium pentobarbital, a craniotomy was carried out for stereotaxic injection. Coordinates used for injection into different brain regions and the injection volumes were summarized in Supplementary Table 1. AAV vectors were stereotaxically injected with a 5-ml Hamilton syringe (65460-02, Replacement needles: 65461-01 or -02) connected to the TJ-4A syringe pump (Longer Precision Pump Co., Ltd.) at a slow flow rate of 0.1 μl min$^{-1}$ to avoid potential damage to local brain tissue. Waited for 15 min after the infusion, then withdrew the needle slowly (300 μm min$^{-1}$) to avoid spilling over.

## Chemogenetic manipulation

Mice with a bilateral infusion of AAV coding for hM4D(Gi) in designated brain regions were used for chemogenetic manipulation of neuronal activities by clozapine-N-oxide (CNO) treatment (3.5 mg kg$^{-1}$, intraperitoneal injection). AAV vectors for the chemogenetic manipulation of neuronal activities include AAV2/9-hSyn-hM4Di-EGFP, AAV2/9-Vglut2-hM4Di-EGFP, AAV2/Retro-hSyn-Cre-EGFP, AAV2/9-hSyn-DIO-GCaMP6s, AAV2/9-hSyn-DIO-mCherry, AAV2/9-hSyn-DIO-hM4Di-mCherry, and AAV2/9-hSyn-DIO-hM3Dq-mCherry were from BrainVTA Co., Ltd (Wuhan, China) or Brain Case Biotech Co., Ltd (Shenzhen, China) (Supplementary Table 1). Behavioral tests were carried out 45 min after the CNO injection. Half the animals were randomly assigned to either the control or treatment groups. All control mice were treated with the same experimental procedures but with the injection of saline instead of CNO.

## Brain cannulation and intracranial administration

Brain cannulation and subsequent intracranial administration were carried out following a previously established protocol[25]. Briefly, five to six weeks following stereotaxic injection of chemogenetic virus vectors, the mice were anesthetized. After opening the skull, a bilateral guide cannula duplex (0.48 mm O.D., 0.34 mm I.D.) (CC 2.0 for LPMR; CC 1.0 for l/vlPAG; RWD, Shenzhen) was carefully inserted into the bilateral LPMR and l/vlPAG regions for CNO infusion. Cannula placements were as follows: for LPMR, AP: −2.1 mm, ML: ±1.0 mm, DV: −2.25 mm; for l/vlPAG, AP: −4.25 mm, ML: ±0.5 mm, DV: −2.05 mm. The guide cannulas were then secured in place with dental cement. To prevent occlusion, a dummy cannula (0.3 mm O.D., CC1.0/2.0; RWD, Shenzhen) was inserted into each guide cannula. The mice were allowed a recovery period of 5 days post-surgery. Bilateral CNO injections (5 μM, 0.5 μl per side) were administered 30 min prior to behavioral tests using an internal injection cannula (0.30 mm O.D., 0.14 mm I.D.) (CC 2.0, G1 0.5 for LPMR; CC 1.0, G1 1.0 for l/vlPAG; RWD, Shenzhen) connected to a microinjection pump (ZS100, Chonry Peristaltic Pump Co., Ltd, China). CNO was injected at a speed of 0.5 μl min$^{-1}$.

## Fiber photometry recording

Four to five weeks following stereotactic surgery, mice injected with AAV2/9-hSyn-DIO-GCaMP6s underwent unilateral implantation of a chronically implanted optical fiber (200 μm diameter, 0.37 numerical aperture; Shanghai Zhuen Bio-Technology Co., Ltd, China). Targeted implantation sites were the vLGN at −2.35 mm bregma, 2.5 mm lateral, −3.6 mm ventral, and the l/vlPAG at −4.25 mm bregma, 0.5 mm lateral, −2.9 mm ventral. Fibers were secured to the skull using miniature screws and dental cement (Shanghai Yuyan Bio-Technology Co., Ltd, China). Animals were allowed to recover on a heating pad post-surgery. An accurate implantation site was confirmed post-experimentally via coronal brain section analysis.

Calcium transients were captured using a fiber photometry system (R810, RWD Life Science Co., Ltd). GCaMP6s fluorescence was excited by a 470 nm LED (30 μW at fiber tip), and calcium-independent signals by a 405 nm LED (20 μW at fiber tip). LEDs alternated at 20 Hz, with emissions recorded by an sCMOS camera (Photo-metrics Prime) at a matching frequency.

In vivo recordings were performed in mice during exposure to open spaces at both ground level and elevated heights. To acclimate them, the mice were handled for 10 min each day across three days. For the tests, each mouse was placed in a transparent cylindrical tube (8.9 cm inner diameter, 20 cm height) set on a round acrylic platform (9.5 cm diameter), located in the center of a 20 × 20 × 20 cm open-field arena with transparent walls. The animal was then exposed to the open space by swiftly removing the tube. Each trial consisted of 30-s recording sessions, separated by 2-min intervals, and was repeated 4–5 times per mouse. The moment of tube removal marked time 0 for each trial. For height exposure trials, the same mice were tested on a similar platform elevated 30 cm above the ground. Signal quantification involved subtracting the calibrated 410 nm signal from the 470 nm signal to correct for both movement and bleaching. ΔF/F ratio was calculated as (470 nm signal - calibrated 410 nm signal) (calibrated 410 nm signal)$^{-1}$, then further normalized using the -5 to 0 s pre-exposure window as the baseline period. The area under the curve (AUC) for the response to the exposure to height or open space was then calculated over a 0 to 10 s period post-exposure.

## Heart rate measurement

A small animal ECG telemetry system (PDT 4000HR E-Mitters system, Starr Life Sciences, USA) was used to measure heart rate in freely moving animals. Under anesthesia, the Mini G2 E-Mitter implant was positioned in the thoracic cavity, angled at 45°–60° to the heart's transverse plane. The device's one wire probe was sutured at the diaphragm, with the other wire probe sutured underneath the

clavicle. Post-implantation, mice were given a two-week recovery period. During heart rate recording sessions, each mouse was placed at the center of the ER-4000 receiver plate located at a high platform and surrounded by transparent walls. The mice were allowed to acclimate in this setup for 1 h. To simulate height exposure, the walls were swiftly removed. The ECG signals were captured by the ER-4000 receiver. Heart rate data was then extracted using VitalView 6 software, which provided readings at a sampling rate of one data point per second over a 30-min period.

### Whole-cell patch-clamp recording

After anesthesia and perfused with artificial cerebrospinal fluid (ACSF) in 4 °C, mouse brains were rapidly removed, and coronal brain slices were cut in an ice-cold oxygenated (95% $O_2$ and 5% $CO_2$) cutting solution (228 mM sucrose, 11 mM glucose, 26 mM $NaHCO_3$, 1 mM $NaH_2PO_4$, 2.5 mM KCl, 7 mM $MgSO_4$, and 0.5 mM $CaCl_2$) at a 300 μm thickness using a vibratome (VT 1200S, Leica Microsystems, Wetzlar, Germany). Slices were recovered in oxygenated artificial cerebrospinal fluid (ASCF) cutting solution at 34 °C for 15 min, followed by ASCF-containing solution (119 mM NaCl, 2.5 mM KCl, 1 mM $NaH_2PO_4$, 1.3 mM $MgSO_4$, 26 mM $NaHCO_3$, 10 mM glucose, and 2.5 mM $CaCl_2$) at 25 ± 1 °C for 1 h before recording. During recording, slices were continuously perfused with the same solution at a rate of 1 ml/min and maintained at 25 ± 1 °C.

Patch pipettes were pulled from borosilicate glass capillary tubes (Cat #64-0793, Warner Instruments, Hamden, CT, USA) using a P-97 pipette puller (Sutter, USA). Whole-cell patch-clamp recordings were carried out on GFP-positive neurons identified under the fluorescence microscope. The internal solution within whole-cell recording pipettes (3–7 MΩ) contained 135 mM potassium gluconate, 7 mM KCl, 10 mM HEPES, 10 mM phosphocreatine, 4 mM ATP-Mg, 0.4 mM GTP-Na (pH 7.2–7.3, 285 mOsm). Patch-clamp recording was performed using a Multiclamp 700B amplifier (Molecular Devices). Recordings from neurons with series resistances >25 MΩ were excluded. Action potentials were recorded under the current-clamp mode. Neurons were held at −60 mV and were injected with different currents (frequency: 0.1 Hz; duration: 500 ms; increments: 20 pA; ranging from −60 to 200 pA). Traces were low-pass filtered at 2 kHz and digitized at 10 kHz (DigiData 1550, Molecular Devices). Data were acquired and analyzed using Clampfit 11.0 software (Molecular Devices). For CNO treatment, brain slices were perfused with ACSF containing 5 μM CNO for 10 min.

### Immunofluorescence

Mice were anesthetized by intraperitoneal administration of sodium pentobarbital (70–80 mg kg⁻¹). After confirmation of anesthesia, mice were perfused with phosphate-buffered saline (PBS; pH 7.6) followed by 4% paraformaldehyde (PFA) in PBS on ice. The brains were post-fixed in 4% PFA for 16–24 h at 4 °C. Subsequently, the brains were sequentially dehydrated by soaking in 20% and 30% sucrose in PBS each for 16–24 h at 4 °C. Serial coronal sections of the brain were cut using a cryostat (Leica CM3050S) at a thickness of 40 μm and processed for standard free-floating immunofluorescence staining. Primary and secondary antibodies used for immunohistochemistry are listed in Supplementary Table 2. These include c-Fos (9F6) rabbit monoclonal antibody (Cell Signaling Technology®, 2250S), used at a dilution of 1:750; c-Fos (2G9C3) mouse monoclonal antibody (Thermo Fisher, MA1-21190), used at 1:1000; anti-Parvalbumin rabbit polyclonal antibody (Abcam, ab11427), at 1:500; and GABA rabbit polyclonal antibody (Sigma, A2052), at 1:500. The secondary antibodies used were goat anti-rabbit IgG H+L (Alexa Fluor® 488, Thermo Fisher, A-11034), goat anti-rabbit IgG H+L (Alexa Fluor® 647, Cell Signaling Technology®, 4414S), goat anti-mouse IgG H+L (Alexa Fluor® 488, Thermo Fisher, A-11029), and goat anti-rabbit IgG H+L (Alexa Fluor® 546, Thermo Fisher, A-11035), all at 1:1000. Sections were transferred onto Super Frost slides and mounted on glass coverslips with mounting media. Fluorescent and bright-field images of brain sections were acquired automatically with the Tissue Cytometry Analysis System (TissueGnostics GmbH, Austria) with a 20× objective.

### Quantification of Fos⁺ cells

Brain sections with equivalent A-P levels from the control and experimental groups were compared. Fluorescent images of brain sections of the target brain region (4–5 sections per brain) were aligned with the coronal reference atlas in Adobe Illustrator CC 1987-2017 22.0.0 (64 bit) to mark the area of each brain region of interest. Fos⁺ cells merged with DAPI in the designated brain region of each brain section were manually counted by an experimenter who was blind to the experiment settings using the Count tool in Adobe Photoshop CC 2018. Total Fos⁺ cells for each brain were determined by adding the cell numbers of all 4–5 sections from this brain.

### Statistical analysis

For most behavioral experiments, data analysis was carried out by an experimenter blind to the experimental conditions whenever possible. Statistical significance was determined using one-tailed or two-tailed Student's *t*-test for single comparisons and One-way ANOVA with post-hoc *t*-tests for multiple comparisons. A $P$ value < 0.05 was considered to be significant. Statistical analysis was conducted using GraphPad Prism 10.2.2 (397) (GraphPad Software, California, USA).

### Reporting summary

Further information on research design is available in the Nature Portfolio Reporting Summary linked to this article.

## Data availability

All data for Figs. 1–6 and Supplementary Figs. 1–12, Movies 1–10, and Tables 1 and 2 are provided in Supplementary Information files. Raw data used to generate Figs. 1–6 and Supplementary Figs. 1–12 were included in Source Data. Source data are provided with this paper. Coordinates and virus used for stereotaxic injection were provided in Supplementary Table 1. Antibodies employed in this study were provided in Supplementary Table 2. Source data are provided with this paper.

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

## Acknowledgements

We thank the ECNU Multifunctional Platform for Innovation (010 and 011) for their mouse breeding service and Dr. Shlomow Wagner and Dr. Shai Netser for their advice in studies of mouse height exposure. We also thank Dr. Haipeng Li for the support in data analysis, and Dr. Huatai Xu and Dr. Dongmin Yin for the studies in the use of Cre lines. This work is supported by the National Key Research and Development Program of China grant 2022YFC2705200 to X.B.Y. and Y.H.P., National Natural Science Foundation of China and Israel Science Foundation Cooperative Research Project grant 32061143016 to X.B.Y. and Y.H.P., and National Natural Science Foundation of China grant 81941013 and 32271022 to X.B.Y. and 31100273 to Y.H.P., Key Laboratory of Brain Functional Genomics at East China Normal University grant 23SKBFGK2 to S.J.W.

## Author contributions

X.B.Y., Y.H.P., S.X., and L.L. conceived the project. S.X., W.S., W.F., Y.S. J.L., and S.J.W. designed and performed the behavioral tests. S.X., W.S., and W.F. conducted chemogenetic manipulations, cortex lesion, whisker shaving, monocular deprivation, and c-Fos+ mapping. J.J. and X.C. performed the whole-cell patch-clamp recordings. W.S. and W.F. conducted immunofluorescence staining. Z.L. and H.S. supported chemical

labyrinthectomy. Y.G. contributed to literature searches and chemogenetic manipulation. Y.H.P., W.S., and W.F. analyzed the data. Y.H.P. prepared figures. X.B.Y. and Y.H.P. supervised the experiments. X.B.Y., Y.H.P., and S.J.W. acquired funding. X.B.Y. wrote the paper with input from all co-authors.

## Competing interests

The authors declare no competing interests.
