## [Peer Review File · Nature Communications]

A non-image-forming visual circuit mediates the innate fear of heights in male miceREVIEWER COMMENTS

Reviewer #1 (Remarks to the Author):

Summary

In this study, Shang, Xie and colleagues investigated neuronal pathways underlying the instinctive fear of heights in mice. They found that instinctive fear of heights primarily relies on visual information, and that other sensory modalities, such as vestibular or somatosensory inputs, only marginally contribute to the effect (Figures 1 and 2). In the subsequent part of the manuscript, the authors investigated the relative contribution of different pathways of the early visual system to this instinctive behaviour. They showed that both the lateral posterior nucleus of the thalamus (LP) and the superior colliculus in the midbrain (SC), areas previously shown to be crucial for other visually guided behaviours, suppress fear of heights (Figure 3), while the primary visual cortex (V1), did not significantly affect the behaviour (Figure 4). Finally, they found that manipulating neurons in the ventral lateral geniculate nucleus (vLGN) and in the periaqueductal gray (PAG) also affected fear of heights (Figures 4 and 5). This study is timely as it addresses an important and physiologically relevant question and adds to the recently growing evidence that the vLGN is a key player in regulating visually guided behaviours.

Many interesting and challenging behaviour manipulation experiments in freely moving mice have been performed and analyzed thoroughly. Overall, the data quality appears good, and results are consistent, however some parts need additional experiments, analyses and/or discussions to be fully convincing and to warrant publication in Nature Communications.

The authors very convincingly show, both by thorough analysis of the behaviour and thanks to the supplementary movies, that mice show a strong and stereotypical aversive response when being exposed to heights (Figure 1). The finding of sensitization after an initial exposure is very interesting and raises many subsequent questions, which I believe should rather be addressed by a follow-up paper. However, some minor clarifications on this point could be helpful (see below).

In the next step, the authors demonstrate that fear of heights predominantly relies on visual cues (Figure 2). The result is strong and convincing. Given the strength of the statement, it could be considered adding more convincing positive controls to prove that vestibular manipulation is working reliably (beyond what is shown in Figure S3) to really avoid any doubts on its efficacy.

In the second half of the manuscript the authors manipulate different brain areas in the early visual system to uncover putative pathways relevant for fear of heights (Figures 3-5). As detailed below, this part is in general less strong and would require more experimental work to be fully convincing. For instance, most manipulation experiments don't rule out alternative possibilities and given that most of the investigated brain areas are heavily interconnected it is so far not clear if the conclusions are fully accurate. Moreover, it is surprising that most of the recent and very relevant vLGN literature, which could support the new findings, has not been mentioned at all. Therefore, some additional experiments and more extensive discussion is needed to make this part fully convincing.

Major

Figures 3 and 4: The results in Figure 3 are believable, given that both SC and LP have been shown very relevant for other visually guided behaviours, such as the mentioned aversive response to visual threat ("loom"). However, it is not clear how the interpretation is consistent with the later results shown in Figure 4 (and Figures S6-S7). Given that LP only (besides the thalamic reticular nucleus) projects to cortical areas (which include V1 and the amygdala), how is it possible that the SC to LP pathway has strong behavioural effects, while none of the downstream targets do? Is the effect mediated through another cortical area or were the manipulations not extensive enough (given the size of cortex) to uncover an effect? Another possibility that should be considered is that the SC to LP pathway could be less involved than another SC pathway (as for example the SC to PAG pathway, which has also been shown crucial for the mentioned looming

behaviour (Evans, Stempel et al., 2018)). This result could still fit the data in Figure 3k-l, when considering branching axons (see point below).

Figures 3k-l and 5i-k: For experiments investigating the importance of a certain pathway (SC to LP and vLGN to PAG) the authors use a similar strategy: retroAAV-cre in region 2, floxed hM4Di in region 1 and systemic injection of CNO. This method has a major caveat, given that it ignores the (likely) possibility that the same cells in region 1, which project to region 2, also project to other regions 3-n. This becomes particularly relevant given that manipulations in LP, SC, vLGN and PAG all showed a behavioural effect and given that it is known that they are all heavily interconnected (vLGN projects directly to LP, SC and PAG. SC projects directly to LP, vLGN and PAG). A cleaner method would be to apply local CNO in region 2 to demonstrate that a particular pathway is involved. Otherwise, these experiments don't add much compared to the single region manipulations. Concluding that the vLGN-PAG circuit mediates the fear of heights (Figure 5) is therefore too strong at this stage. Also, how does it fit with the SC to LP pathway that also mediates the fear of heights (Figure 3)?

Figures 3-5: The authors always manipulate brain activity by silencing with hM4Di. Do they believe that effects are bidirectional as it has been shown for many behaviours controlled by the vLGN?

General: This study concludes with the importance of the vLGN to PAG pathway to mediate fear of heights. While it is not fully clear why the authors focused on this pathway versus the SC to LP pathway for example, which shows similarly strong effects, it is still interesting that non-classical visual pathways appear to be involved. However, given that vLGN has historically been understudied, it is surprising that most new papers showing vLGN's strong involvement in many instinctive behaviours (as reviewed recently in Fratzl and Hofer, 2022) have been completely ignored. For instance, the authors acknowledge that the SC and the LP have been shown relevant for another visually guided defensive behaviour ("looming"), while they did not mention that two independent groups (Salay and Huberman, 2021, Fratzl et al., 2021) found that different vLGN pathways are also critical for this same behaviour. For instance, both groups found that vLGN inhibition increased the amount of defensive behaviour displayed, which is the opposite of what the authors found here. These results should be discussed accordingly. In general, also given the interconnectivity of the studied subcortical areas (see previous points), the paper needs a significantly more thorough discussion putting into perspective how these different brain areas, including the vLGN, could act together to mediate fear of heights.

Minor

Figures 1i,j and S2: Is there also a cross-modality effect in the fear facilitation upon re-exposure? Between gray/transparent walls and open walls (or vice versa)?

Figure 4m,n: The effect size for vLGN manipulation appears relatively weak and is likely significant only because of the very large n (also compared to other experiments). Can this be explained by poor expression of the construct (as shown in Figure 4m)? Or is the argument that the subpopulation that projects to PAG has a stronger effect? There seems to be many more cells labelled in Figure 5i than Figure 4m, so this argument would not be believable without additional work. Also, there appears to be quite some labelling in layer 2/3 of the cortex in the right hemisphere (Figure 4m), which is very surprising given that there is no trivial explanation why it should be labelled (axons should not project there). Could the authors clarify this point? Moreover, the figure label of the construct ("rAVV") also appears wrong. Is this a different construct than the one used in V1 (Figure 4k)? In general, labels for constructs appear inconsistent across figures (for instance hM4Di vs hM4D(gi)) and would be best if they would be the same for identical constructs. Figure 4m,n: It appears a bit odd that Figure 4m,n is still part of Figure 4 ("The primary visual cortex is dispensable for fear of heights"). This part is one of the main findings regarding vLGN's involvement in the behaviour and is logically not really connected to the V1 experiments. Same comment for the corresponding parts in the text.

Figure S6: The authors show that vLGN inhibition has no effect on animals in the open field test. This result is surprising given that both Salay and Huberman, 2021 and Fratzl et al., 2021

independently showed very strong effects of vLGN manipulation in exactly this behaviour. The authors should discuss possible differences in their approach.

Discussion: "We made an intriguing discovery that neither V1 nor SC is necessary for the physiological fear of heights in naïve mice." Should SC be amygdala instead? The SC is involved in the behaviour as shown in the paragraph just before. Moreover, it could be consider mentioning Evans, Stempel et al., 2018, who also found that V1 and amygdala manipulations did not alter the "looming" behaviour, which would be consistent for an instinctive visually guided defensive behaviour (see Extended Data Fig. 3).

Reviewer #2 (Remarks to the Author):

There have been limited studies utilizing mice to investigate the neural circuitry underlying acrophobia. This study examines the specific involvement of the SC-LP and LGN-I/vIPAG pathway in acrophobia in mice, offering potential insights into the underlying mechanisms. However, several major issues need to be addressed before the full impact of this manuscript can be determined.

1. The definition of trembling behavior in mice should be clarified, as it may not be clearly observable on the video. It would be helpful to provide references supporting the chosen behavioral paradigm.
2. In addition to the observed increases in c-Fos expression in SC, LP, and vLGN after exposure to height, it would be interesting to explore whether other brain regions, such as PBGN or VTA, also show changes. In addition, real-time monitoring of neural activity in these brain areas throughout the entire fear of heights experiment would provide valuable insights into their activation during different stages.
3. The rationale for using a concentration of 3.5 mg/kg of CNO should be explained, as previous studies have shown that 0.5-2 mg/kg is effective.
4. The authors states that the inhibition of the SC-LPMR pathway has presynaptic effects only, but it would be important to investigate whether inhibition of postsynaptic LPMR neurons produces similar effects.
5. Considering that PV-positive neurons in SC can project to LPTN, it would be informative to compare the SC neurons projecting to LPMR and LPTN. Are there distinct subpopulations of SC neurons projecting to LPMR? How do these different populations of SC neurons become activated in response to looming or fear of heights?
6. It would be interesting to investigate whether direct activation of the SC-LPMR pathway can reduce fear. Additionally, can activation of the vLGN-PAG pathway induce trembling under conditions of height that do not induce fear?
7. The quality of the fluorescence images of the brain slices, especially the c-Fos images induced by height, could be improved. Furthermore, inconsistencies in the scale and size of the brain slice images within the same brain area should be addressed. The scale bars in some of the brain slice images also appear to be incorrect.
8. Since the fear of heights in mice is more pronounced on the 2nd or 7th day, it would be interesting to investigate whether modulation of the two circuits during this period could block this fear. Are there any plasticity changes in the relevant brain areas?
9. Considering that vLGN also projects to SC (Fratzl et al., 2021), how does this relate to its connection with the SC-LP and LGN-SC pathway? How does this interplay contribute to the fear of heights in mice under physiological conditions?

Reviewer #3 (Remarks to the Author):

The authors investigate aversive behaviour to height stimuli in mice in order to obtain clues as to the neurophysiological background of the height vertigo-fear of height phenomenon.

The article is quite informative and there is definitely a lot of work behind it. The authors first phenotype the mouse-specific response to a height stimulus at the behavioural level. Then they quite systematically manipulate environmental stimulus conditions on the one hand and stimulus perception conditions on the other. In particular, they work out the central role of visual stimuli. In the next step, they identify components along the various neural visual processing networks that respond specifically to the height stimulus. In a final step, they selectively switch off the function of these components chemically or surgically and, with regard to potential changes at the behavioural level, are able to elaborate two visual circuits that very specifically link visual height stimuli and fear response. The writing style of the manuscript is very technical, but this is quite common for articles published by Nature Communications.

Major queries:

- (1) In several places, the terms concerning physiological and non-physiological responses to altitude exposure are not used correctly (Introduction, Discussion). In addition to a physiological response to altitude exposure affecting 100% of the population, there is a visual height intolerance that occurs in about 30% of the population, with a continuum to acrophobia affecting about 6% of the population (Brandt and Huppert, *Curr. Opin., Neurol.*, 2014)
- (2) Fear of heights is described as a motion sickness ('Section 'vestibular input is dispensable...'; 2nd paragraph Discussion). This is not correct.
- (3) Generally speaking, the presentation of the results in the various subsections is very detailed, sometimes difficult to read. In this respect, a summary table on magnetic resonance imaging changes would be useful. In the Discussion section at the end, the magnetic resonance imaging changes are again presented in great detail. A less detailed discussion style would make more sense here.
- (4) It is not entirely convincing that the quantified behavioural changes at altitude should be specific to fear of heights. For that, they would have to be contrasted with behavioural reactions to other fear stimuli, which the authors do not do. This aspect also fundamentally affects the other network analyses. There, too, the current methodological approach cannot convincingly show that the response is specific to a height stimulus and does not occur with other fear-inducing stimuli. Moreover, the description of how the individual behavioural parameters were quantified is only very cursory; as a reader, I would like to have more background information. I also wonder why autonomic responses (e.g. pulse increase) were not also measured; in earlier studies cited by the authors and in which they were also partly involved, ECG measurements were carried out in mice at altitude; this seems to be an established and common methodology.
- (5) The authors report the height-dependent saturation of fear of heights, similar to what Brandt described in the 1980s. In humans, virtual simulation was used to test the dependency of neurophysiological responses to different heights, where saturation was found (Wuehr et al., *J. Neurol.*, 2019). However, this aspect is not taken up and discussed further by the authors. It would be interesting to see how this phenomenon can be explained in the context of the new findings and the authors' clear rejection of the sensory mismatch theory.
- (6) The authors examine the effect of repeated altitude exposure on the behavioural level and report an increased fear response to repeated altitude stimulation. This contradicts data in the literature, where a rather constant reaction could be found, and other reports from behavioural therapy. The authors should definitely discuss this aspect more, because it calls into question whether the animal model is really a suitable model for human behavioural response.
- (7) The exposure to heights in the dark is difficult to understand. Overall, little detail is given, e.g. whether the mice were naïve or were able to explore the platform in the light beforehand; and if they were naïve, whether many mice then simply walked over the edge of the platform.
- (8) Overall, the role of the individual behavioural tests (open field, elevated open maze, etc.) and the implications drawn from the test results are not entirely clear. The information given in the manuscript about this is very sparse. For example, why is the open field test an indicator of general anxiety levels? At first glance, this does not seem entirely plausible.
- (9) Some abbreviations in the text are not introduced, for example CeA.

Reviewer #1 (Remarks to the Author):

Summary

In this study, Shang, Xie and colleagues investigated neuronal pathways underlying the instinctive fear of heights in mice. They found that instinctive fear of heights primarily relies on visual information, and that other sensory modalities, such as vestibular or somatosensory inputs, only marginally contribute to the effect (Figures 1 and 2). In the subsequent part of the manuscript, the authors investigated the relative contribution of different pathways of the early visual system to this instinctive behaviour. They showed that both the lateral posterior nucleus of the thalamus (LP) and the superior colliculus in the midbrain (SC), areas previously shown to be crucial for other visually guided behaviours, suppress fear of heights (Figure 3), while the primary visual cortex (VI), did not significantly affect the behaviour (Figure 4). Finally, they found that manipulating neurons in the ventral lateral geniculate nucleus (vLGN) and in the periaqueductal gray (PAG) also affected fear of heights (Figures 4 and 5). This study is timely as it addresses an important and physiologically relevant question and adds to the recently growing evidence that the vLGN is a key player in regulating visually guided behaviours.

Many interesting and challenging behaviour manipulation experiments in freely moving mice have been performed and analyzed thoroughly. Overall, the data quality appears good, and results are consistent, however some parts need additional experiments, analyses and/or discussions to be fully convincing and to warrant publication in Nature Communications.

The authors very convincingly show, both by thorough analysis of the behaviour and thanks to the supplementary movies, that mice show a strong and stereotypical aversive response when being exposed to heights (Figure 1). The finding of sensitization after an initial exposure is very interesting and raises many subsequent questions, which I believe should rather be addressed by a follow-up paper. However, some minor clarifications on this point could be helpful (see below).

In the next step, the authors demonstrate that fear of heights predominantly relies on visual cues (Figure 2). The result is strong and convincing. Given the strength of the statement, it could be considered adding more convincing positive controls to prove that vestibular manipulation is working reliably (beyond what is shown in Figure S3) to really avoid any doubts on its efficacy.

In the second half of the manuscript the authors manipulate different brain areas in the early visual system to uncover putative pathways relevant for fear of heights (Figures 3-5). As detailed below, this part is in general less strong and would require more experimental work to be fully convincing. For instance, most manipulation experiments don't rule out alternative possibilities and given that most of the investigated brain areas are heavily interconnected it is so far not clear if the conclusions are fully accurate. Moreover, it is surprising that most of the recent and very relevant vLGN literature, which could support the new findings, has not been mentioned at all. Therefore, some additional experiments and more extensive discussion is needed to make this part fully convincing.

➡ Thank you for acknowledging the strength of our results. We appreciate your suggestion to further substantiate the effectiveness of our vestibular manipulations with additional positive controls. In response, we have conducted vestibulo-ocular reflex (VOR) assays, a well-established and reliable technique for assessing vestibular function. These assays objectively measure vestibular integrity by evaluating reflexive eye movements in response to head movements. The results of these assays, which we have included in the revised manuscript (Supplementary Fig. 4g), demonstrate a clear functional effect of bilateral vestibular manipulations. In cases of unilateral vestibular manipulation, we observed spontaneous nystagmus and tail suspension swing (Supplementary Movie 8), both indicative of vestibular imbalance. These new results, added to the revised manuscript, provide further support to the validity of our vestibular manipulation methods.

Major

Figures 3 and 4: The results in Figure 3 are believable, given that both SC and LP have been shown very relevant for other visually guided behaviours, such as the mentioned aversive response to visual threat ("loom"). However, it is not clear how the interpretation is consistent with the later results shown in Figure 4 (and Figures S6-S7). Given that LP only (besides the thalamic reticular nucleus) projects to cortical areas (which include V1 and the amygdala), how is it possible that the SC to LP pathway has strong behavioural effects, while none of the downstream targets do? Is the effect mediated through another cortical area or were the manipulations not extensive enough (given the size of cortex) to uncover an effect? Another possibility that should be considered is that the SC to LP pathway could be less involved than another SC pathway (as for example the SC to PAG pathway, which has also been shown crucial for the mentioned looming behaviour (Evans, Stempel et al., 2018)). This result could still fit the data in Figure 3k-l, when considering branching axons

(see point below).

➔ Thank you for your insightful comments and the opportunity to discuss the results relevant to the role of the SC to LP pathway in fear of heights.

Alignment of results across figures: We appreciate your concern regarding the consistency of interpretations between Fig. 3 and 4. However, we believe there is no discrepancy between these results. Specifically, our results show that chemogenetic inhibition of one important downstream target of LP, specifically the basal/lateral amygdala (BLA), yields behavioral effects akin to the inhibition of either LP or SC independently (the revised Supplementary Fig. 11). This consistency reinforces our hypothesis that the SC-LP pathway, potentially via the BLA, plays a pivotal role in modulating the defensive response to height threat. Given LP's extensive cortical projections, we also acknowledge the potential involvement of other cortical areas beyond the visual cortex. We have expanded on this discussion in the revised manuscript (P. 12, line 31 to P. 13, line 15).

SC-PAG pathway and fear of Heights: Regarding the potential role of the SC to the dorsal periaqueductal gray (PAG) pathway in mediating fear of heights, our study suggests an alternative mechanism. We observed that exposure to heights activates neurons in the SC, and inhibiting SC activity increases the fear of heights. This suggests that SC activation in response to height stimuli may suppress, rather than provoke, the defensive response to height threats. Considering the PAG's established involvement in fear and anxiety responses, our findings imply that the SC-PAG pathway may not be directly responsible for the fear response to heights. However, exploring the regulatory influence of the SC-PAG pathway on height-induced fear is a promising direction for future research. We have included further discussion on this point in the revised manuscript (P. 15, lines 18-23).

Figures 3k-l and 5i-k: For experiments investigating the importance of a certain pathway (SC to LP and vLGN to PAG) the authors use a similar strategy: retroAAV-cre in region 2, floxed hM4Di in region 1 and systemic injection of CNO. This method has a major caveat, given that it ignores the (likely) possibility that the same cells in region 1, which project to region 2, also project to other regions 3-n. This becomes particularly relevant given that manipulations in LP, SC, vLGN and PAG all showed a behavioural effect and given that it is known that they are all heavily interconnected (vLGN projects directly to LP, SC and PAG. SC projects directly to LP, vLGN and PAG). A cleaner method would be to apply local CNO in region 2 to demonstrate that a particular pathway is involved. Otherwise, these experiments don't add much compared to the single region manipulations. Concluding that the vLGN-PAG circuit mediates the fear of heights (Figure 5) is therefore too strong at this stage. Also, how

does it fit with the SC to LP pathway that also mediates the fear of heights (Figure 3)?

➔ Thank you for your insightful critique regarding the methodology used in our study and for highlighting the potential issue of multi-target effects in our methodology. We appreciate the suggestion of local application of CNO at the downstream brain area to address the function of a particular pathway. In response to your suggestion, we have conducted additional experiments where we applied CNO locally to LPMR and l/vIPAG to specifically inhibit the downstream terminals of the SC-LP projection and the vLGN-PAG projection. These targeted experiments were designed to assess the role of these specific pathways more directly. Consistent with previous findings, results from these additional experiments (the revised Fig. 3m, n and Fig. 6o, p) were generally consistent with those observed from the inhibition of SC and vLGN alone. These findings further corroborate our hypothesis that the SC-LP circuit suppresses, while the vLGN-PAG circuit promotes, the defensive response elicited by the threat of heights.

While these findings reinforce the role of the specific circuits we investigated, we acknowledge that the involvement of other downstream targets in the fear of heights cannot be completely ruled out at this stage. The extensive interconnections among these key brain regions imply potential complex interactions between these pathways to fine-tune the animal's behavioral response to height stimuli in varying environmental contexts. We have included further discussion on this point in the revised manuscript (P. 13, lines 8-15; P. 15, line 24 to P. 16, line 2).

Regarding your query about the relationship between the SC-LP and vLGN-PAG pathways in mediating fear of heights, our current understanding suggests that these are parallel pathways activated upon exposure to heights, with opposing behavioral consequences. Given the substantial interconnections among these brain areas, such as SC projecting to PAG and vLGN projecting to SC and LPTN, there likely exists a level of crosstalk between these two parallel pathways to fine-tune the animal's behavioral response to height stimuli. The vLGN's projections to the SC and LPTN are particularly noteworthy. Given that most of the vLGN's output neurons are GABAergic, it is plausible that these GABAergic projections provide feedforward inhibition to the SC-LP pathway. This inhibition could potentially dampen the SC-LP pathway's suppression of height fear, thereby amplifying the fear response to height threat. Future investigations will be crucial to unravel these complex interactions and fully elucidate their roles in height-induced fear responses. We have included this discussion in the revised manuscript (P. 15, line 30 to P. 16, line 2).

Figures 3-5: The authors always manipulate brain activity by silencing with hM4Di. Do they believe that effects are bidirectional as it has been shown for many

behaviours controlled by the vLGN?

➔ Thank you for highlighting the crucial aspect of bidirectionality in neuron and circuit manipulation. We acknowledge the importance of examining both inhibitory and excitatory effects to fully understand the functions of these circuits. Following your suggestion, we conducted additional experiments to investigate the effects of excitatory modulation in the SC, vLGN, and PAG on the fear of heights, using hM3Dq-mediated chemogenetic activation.

SC excitation: Our preliminary findings indicate that chemogenetic activation of SC prompts a "flee" behavior in mice, leading them to ignore height threats and rapidly escape when placed on top of a high platform, as demonstrated in Video_1_for reviewers only: Abnormal mouse behavior post pan-activation of the SC. While this aligns with the role of SC in mediating escape behavior in response to visual threats and supports the notion of SC's involvement in suppressing height-induced fear, it is worth noting that such intense activation may not reflect normal brain activity. Due to the potential non-physiological nature of this response, we have chosen not to include these results in the revised manuscript. In the future, we plan to refine our approach, particularly by adjusting the CNO dosage and by a more targeted activation of the specific neuronal ensemble within the SC. Through these improvements, we hope to gain a more accurate understanding of the SC's role in modulating the fear of heights and its overall impact on behavior in the future.

PAG excitation: Our experiments showed that chemogenetic activation of the l/vIPAG results in a freezing-like behavior in mice, as shown in the Fig. 6n and Supplementary Movie 10. This finding corroborates the established role of the PAG in governing fear and anxiety responses. Similar to our approach with the SC excitation experiment, refinements are necessary to enhance the physiological relevance of this intervention. This includes fine-tuning the dosage of CNO and focusing on the activation of specific neuronal ensembles within the l/vIPAG that are responsive to height.

vLGN excitation: Our preliminary experiment showed that activation of vLGN neurons led to seizure-like behaviors in mice on the open high platform (OHP), Video 2_for_reviewers only: Abnormal mouse behavior post pan-activation of the vLGN, suggesting strong fear and anxiety responses. However, this too may represent an overstimulation rather than a physiological reaction.

In conclusion, our preliminary findings, while hinting at potential behavioral outcomes of circuit excitation, underscore the complexity of these responses and the necessity for more detailed investigation. We hope to conduct more careful studies in the future to achieve a more physiologically relevant excitation of these brain areas and circuits to assess its effect on fear of heights.

General: This study concludes with the importance of the vLGN to PAG pathway to mediate fear of heights. While it is not fully clear why the authors focused on this pathway versus the SC to LP pathway for example, which shows similarly strong effects, it is still interesting that non-classical visual pathways appear to be involved. However, given that vLGN has historically been understudied, it is surprising that most new papers showing vLGN's strong involvement in many instinctive behaviours (as reviewed recently in Fratzl and Hofer, 2022) have been completely ignored. For instance, the authors acknowledge that the SC and the LP have been shown relevant for another visually guided defensive behaviour ("looming"), while they did not mention that two independent groups (Salay and Huberman, 2021, Fratzl et al., 2021) found that different vLGN pathways are also critical for this same behaviour. For instance, both groups found that vLGN inhibition increased the amount of defensive behaviour displayed, which is the opposite of what the authors found here. These results should be discussed accordingly. In general, also given the interconnectivity of the studied subcortical areas (see previous points), the paper needs a significantly more thorough discussion putting into perspective how these different brain areas, including the vLGN, could act together to mediate fear of heights.

➡ Thank you for your valuable comments. We appreciate the opportunity to further clarify our research focus and discuss the broader context of our findings.

The focus on the vLNG versus the SC: Our focus on the vLGN was driven by our findings indicating a positive influence of the vLGN in fear of heights. While we recognize the significant effects of the SC to LP pathway in visual threat response, we found that inhibition of either SC or LP elevated the anxiety level of mice on the OHP (Fig. 3). Therefore, the SC to LP pathway is more likely to play a regulatory role upon height threat, preventing the over-reaction of the animal under such threat. In contrast, inhibition of vLGN significantly reduced the fear and anxiety of mice on the OHP, suggesting an important role of vLGN in mediating the defensive response to height threat. That's why we focused on the role of vLGN and its downstream target PAG in fear of heights.

Citation of recent literatures: Thank you for highlighting our oversight in not fully discussing the extensive role of the vLGN in various brain functions and its involvement in numerous instinctive behaviors, as comprehensively reviewed in the work of Fratzl and Hofer (2022). Accordingly, we have updated our manuscript to include a more detailed discussion to ensure that our findings are interpreted within a broader and more informed context, highlighting the multifaceted role of the vLGN in neural processing and behavior (P. 14, line 19 to P. 15, line 6).

Discussion on brain area interconnectivity: Thank you for highlighting the

interconnectivity of different brain areas related to fear of heights, including the vLGN, SC, and their upstream and downstream brain regions. In response to this suggestion, we expanded our discussion on the role of vLGN and its potential interplay with other brain regions in mediating fear of heights (P. 14, line 19 to P.15, line 6; P. 15, line 24 to P.16, line 2).

Minor

Figures 1i,j and S2: Is there also a cross-modality effect in the fear facilitation upon re-exposure? Between gray/transparent walls and open walls (or vice versa)?

➡ Thank you for your insightful question regarding the potential cross-modality effects in the facilitation of fear of heights following prior exposure. In response to your suggestion, we conducted additional experiments to explore how different initial exposure modalities might influence mouse behavior during following re-exposure. In these experiments, naïve mice were first exposed to either an OHP, a high platform with transparent walls (TWP), or a high platform with non-transparent, gray walls (GWP) (Trial 1, T1). After a two-day interval, these mice were then re-exposed to either the same or a different platform condition (Trial 2, T2). We assessed their anxiety levels during both exposures by analyzing behaviors such as locomotion patterns, trembling, grooming, and rearing up durations. Our results, depicted in the Fig. 1_for_reviewers (next page), particularly highlight the O→T scenario (T1 (OHP) followed by T2 (TWP)). Mice in this group displayed signs of fear and anxiety during the TWP re-exposure (T2 (TWP)), which were not observed in mice directly exposed to TWP (the T1(TWP) group). However, the anxiety level in the T2 of O→T scenario was not as pronounced as it was in mice directly exposed to OHP (the T1 (OHP) group). Considering that the transparent walls enable the animal to perceive the association of the environment with a height threat, this increased anxiety during T2 (TWP) suggests that prior exposure to OHP may sensitize mice, increasing their nervousness in an environment associated with previous height threats. Such a response could be akin to the contextual aspect of fear conditioning. Given the complexity of these findings and the need for extensive future research to further clarify this potential mechanism, we have chosen to discuss this aspect with the reviewers but not include it in the main body of the revised manuscript.

Fig. 1_for_reviewers

Figure 4m,n: The effect size for vLGN manipulation appears relatively weak and is likely significant only because of the very large n (also compared to other experiments). Can this be explained by poor expression of the construct (as shown in Figure 4m)? Or is the argument that the subpopulation that projects to PAG has a stronger effect? There seems to be many more cells labelled in Figure 5i than Figure 4m, so this argument would not be believable without additional work. Also, there appears to be quite some labelling in layer 2/3 of the cortex in the right hemisphere (Figure 4m), which is very surprising given that there is no trivial explanation why it should be labelled (axons should not project there). Could the authors clarify this point? Moreover, the figure label of the construct (“rAVV”) also appears wrong. Is this a different construct than the one used in V1 (Figure 4k)? In general, labels for constructs appear inconsistent across figures (for instance hM4Di vs hM4D(gi)) and would be best if they would be the same for identical constructs.

➡ **Potential reason for a modest effect:** Thank you for your detailed observations. First, the relatively large sample size for this data results from conducting multiple

repeated experiments to ensure the reproducibility of this critical finding. We acknowledge your point regarding the modest effect observed following vLGN inhibition. One potential explanation for this could be the limited expression of the viral vector in the relatively small vLGN nucleus, indicating a need for improvement in the efficiency of viral vector delivery to this specific area. Additionally, the vLGN is composed of a mix of GABAergic neurons and a smaller proportion of glutamatergic neurons. The chemogenetic inhibition approach we employed was non-selective, impacting both neuronal populations. This non-specific inhibition may have diluted the functional impact of vLGN inhibition, leading to the less pronounced behavioral changes observed compared to those following PAG inhibition.

In response to your comment, we have conducted additional analyses to affirm the significance of the vLGN-PAG pathway in the fear of heights. We examined the effects of chemogenetically inhibiting the terminals of this pathway in the PAG. These further experiments align with our initial results from the chemogenetic inhibition of vLGN neurons. The outcomes demonstrated a notable decrease in the defensive responses of mice to height threats, evidenced by increased exploratory behavior and a reduction in trembling duration on the OHP. These new findings are presented in the updated Fig. 6o, p of our revised manuscript, providing additional support for the role of the vLGN-PAG pathway in height-induced fear responses.

Comparison with PAG inhibition: In contrast to vLGN inhibition, chemogenetic inhibition of the PAG showed a more substantial impact, nearly eliminating the fear of heights in most animals. This suggests that, while vLGN plays an important role in height fear, it might operate in conjunction with other visual pathways. These pathways could converge at the PAG, synergistically contributing to the fear and anxiety response.

Addressing the unexpected labeling: Regarding the unexpected labeling in layer 2/3 of the cortex in the right hemisphere in the revised Fig. 5l, we attribute this to inadvertent leakage of the viral vector during microinjection. We have since optimized our microinjection protocol to minimize such spillage and updated the figure accordingly.

Consistency in labeling of virus constructs: We have also addressed the inconsistencies in labeling the viral constructs across different figures. The errors in labeling, such as “rAVV” and the varying designations for hM4Di, have been corrected in the revised manuscript for clarity and accuracy.

Figure 4m,n: It appears a bit odd that Figure 4m,n is still part of Figure 4 (“The primary visual cortex is dispensable for fear of heights”). This part is one of the main findings regarding vLGN’s involvement in the behaviour and is logically not really

connected to the V1 experiments. Same comment for the corresponding parts in the text.

➡ Thank you for your insightful comment regarding the organization of the original Figure 4m, n. We agree that the placement of these figures within the section titled “The primary visual cortex is dispensable for fear of heights” may not have logically connected with the main findings related to the vLGN’s involvement in the behavior.

In light of your valuable feedback, we have revised the organization of our figures to better align with the logical flow of our results. The data specifically pertaining to the vLGN have been separated from those related to V1. This reorganization can be seen in the newly created Fig. 5 in the revised manuscript (P. 8, lines 16-34; P. 9, lines 1-9). We believe this adjustment enhances the clarity and coherence of our narrative, allowing readers to follow the progression of our findings and their implications more easily.

Figure S6: The authors show that vLGN inhibition has no effect on animals in the open field test. This result is surprising given that both Salay and Huberman, 2021 and Fratzl et al., 2021 independently showed very strong effects of vLGN manipulation in exactly this behaviour. The authors should discuss possible differences in their approach.

➡ Thank you for highlighting the discrepancy between our findings in the open field test following vLGN inhibition and the results reported by Salay and Huberman (2021) and Fratzl et al. (2021). We value this observation and have carefully examined potential reasons for these differing outcomes.

The observed discrepancy between our findings in the open field test and those reported by Salay and Huberman (2021) and Fratzl et al. (2021) may indeed be attributed to key differences in experimental design and methodologies. In our study, we employed a pan-neuronal inhibition approach. This contrasts with the methodologies of Salay and Huberman, and Fratzl et al., who focused specifically on inhibiting GABAergic neurons within the vLGN. Such a difference in the targeted neuronal populations could be a likely factor contributing to the variation in behavioral outcomes observed in the open field test. Moreover, the duration of the open field test in their studies was limited to 5 minutes, whereas in our experiments, we extended the observation period to 30 minutes. This longer duration of behavioral assessment in our study could have also influenced the outcomes.

In light of your valuable comment, we have expanded our study to include chemogenetic inhibition of the GABAergic neurons in the vLGN. Our findings revealed that this treatment led to a significant reduction in the fear and anxiety of mice on the OHP, as shown in the revised Fig. 5n, o. This effect mirrors the results

observed following the pan-neuronal inhibition of the vLGN.

Further, in the open field test, we observed a trend of decreased exploratory behavior in mice following the chemogenetic inhibition of the vLGN's GABAergic neurons (please refer to next page, Fig. 2_for_reviewers). Although this trend did not reach statistical significance, likely due to the limited sample size of this experiment, it is consistent with the results of prior studies by Salay and Huberman, and Fratzl et al., which showed a significant reduction in central exploration in their open field test after inhibiting GABAergic neurons in the vLGN.

Given the lack of conclusive statistical power in the open field test results, we have chosen not to include this piece of finding in our revised manuscript. Nevertheless, we have expanded our discussion of these observations, placing them in the context of existing literature (P. 14, line 19 to P. 15, line 6).

Fig. 2_for_reviewers

Discussion: "We made an intriguing discovery that neither V1 nor SC is necessary for the physiological fear of heights in naïve mice." Should SC be amygdala instead? The SC is involved in the behaviour as shown in the paragraph just before. Moreover, it could be consider mentioning Evans, Stempel et al., 2018, who also found that V1 and amygdala manipulations did not alter the "looming" behaviour, which would be consistent for an instinctive visually guided defensive behaviour (see Extended Data Fig. 3).

➡ Thank you for your constructive comment and for pointing out the potential ambiguity in our discussion regarding the roles of the SC and V1 in the physiological fear of heights in naïve mice. We recognize the need to clarify this aspect, as our statement might have led to some misunderstanding of our findings.

Our findings suggest that the SC plays a role in modulating or suppressing the fear of heights, rather than being a direct mediator of this fear. This distinction is crucial; while the SC is involved in the behavioral response to heights, it is not indispensable for the initial manifestation of the physiological fear of heights in naïve

mice. This is in line with our initial statement that both the SC and V1 are not necessary for this innate fear response.

We agree that your suggestion to consider the findings of Evans, Stempel et al., 2018, is valuable. Their study, which showed that manipulations in the V1 and amygdala did not block the "looming" behavior, supports the idea of instinctive visually guided defensive behaviors being mediated through mechanisms other than direct involvement of V1 or amygdala. This aligns well with our findings, suggesting that these regions, including the V1 and amygdala, may play more of a modulatory role in processing instinctive visually guided defensive behaviors.

In light of your feedback, we have revised the discussion section to better articulate the regulatory role of the SC in fear of heights (P. 13, lines 17-30). Additionally, we have incorporated a discussion on the findings of Evans, Stempel et al., 2018, to contextualize our results within the broader scope of existing research on instinctive fear responses (P. 14, lines 1-3).

Reviewer #2 (Remarks to the Author):

There have been limited studies utilizing mice to investigate the neural circuitry underlying acrophobia. This study examines the specific involvement of the SC-LP and LGN-l/vIPAG pathway in acrophobia in mice, offering potential insights into the underlying mechanisms. However, several major issues need to be addressed before the full impact of this manuscript can be determined.

1. The definition of trembling behavior in mice should be clarified, as it may not be clearly observable on the video. It would be helpful to provide references supporting the chosen behavioral paradigm.

➡ Thank you for highlighting the need for a more precise definition of trembling behavior in mice within our study. We acknowledge the challenge in discerning this behavior without established references. In our research, we define trembling behavior as the absence of locomotion combined with visible upper body fluctuations, frequently observed when mice are exposed to height threats. This is distinct from the well-documented freezing behavior, which is characterized by an immobilization of skeletal muscles but does not include cessation of internal functions like heartbeats. Freezing is commonly observed in classical fear conditioning paradigms and in response to "looming" stimuli.

Notably, while freezing behavior typically involves a complete immobilization of the whole-body musculature, trembling, as we have observed, seems to be associated

with an inhibition of limb movements but not necessarily of muscles in the head and neck area. To better illustrate this, we have included additional video content in our revised manuscript. These videos showcase trembling episodes as mice explore the edge of a high platform, in contrast to freezing behavior induced by contextual fear conditioning.

These clarifications and additional video evidence have been incorporated into the revised manuscript to provide a more comprehensive understanding of the observed trembling behavior in mice (P. 3, lines 15-21; Supplementary Movie 2).

2. In addition to the observed increases in c-Fos expression in SC, LP, and vLGN after exposure to height, it would be interesting to explore whether other brain regions, such as PBGN or VTA, also show changes. In addition, real-time monitoring of neural activity in these brain areas throughout the entire fear of heights experiment would provide valuable insights into their activation during different stages.

➡ Thank you for your insightful suggestion to explore the involvement of additional brain regions, such as the PBGN and VTA, and to utilize real-time neural activity monitoring in our study. We have conducted additional experiments accordingly. Findings of these additional experiments have significantly enriched our study, providing a deeper understanding of the neural mechanisms involved in fear of heights.

Following your recommendation, we extended our analysis to include other brain regions implicated in defensive and emotional responses, namely parabrachial nucleus (PBGN), ventral tegmental area (VTA), ventromedial hypothalamus (VMH), and ACC (anterior cingulate cortex). Using c-Fos immunofluorescence staining, our findings indicate minimal activity in PBGN in both the presence and absence of height stimuli (P. 7, lines 6-9; Fig. 3_for_reviewers on next page), suggesting its limited involvement in fear of heights. Conversely, ACC, VMH, and VTA showed increased Fos signal in response to height stimuli. This heightened activity suggests their potential roles in the arousal and cognitive processing of height threats, contributing to the animals' complex behavioral responses. We have elaborated on the implications of these findings in our revised manuscript (P. 16, lines 3-8; Supplementary Fig. 12).

In line with your suggestion for real-time neural activity monitoring, we performed calcium imaging to observe the activities of GABAergic neurons in vLGN (Fig. 5f-k) and glutamatergic neurons in l/vIPAG (Fig. 6d-g) in response to height stimulus. This approach revealed that these neurons are indeed activated upon height exposure, corroborating our c-Fos staining results.

Fig. 3_for_reviewers

PBGN c-fos mapping

3. *The rationale for using a concentration of 3.5 mg/kg of CNO should be explained, as previous studies have shown that 0.5-2 mg/kg is effective.*

➡ Thank you for your question regarding our choice of CNO concentration in our experiments. We acknowledge that previous studies have indeed reported effective chemogenetic activation or inhibition with CNO dosages ranging from 0.5 to 2 mg/kg. However, our decision to use a higher dosage of 3.5 mg/kg was carefully considered and is supported by two main factors:

Our review of recent high-profile research articles revealed instances where CNO concentrations as high as 5 mg/kg were used for chemogenetic inhibition (Salay and Huberman, *Cell Report*, 2021; Fratzl et al., *Neuron*, 2021). These studies provided a basis for considering a higher dosage in our experiments.

Our own empirical observations have shown that a CNO dosage of 3.5 mg/kg effectively suppresses c-fos signal in multiple brain regions, as demonstrated in Supplementary Fig. 5c, d of the revised manuscript. This dosage was found to be sufficient for the purposes of chemogenetic inhibition in our specific experimental setup.

We understand that the optimal dosage of CNO depends on various factors, including the efficiency of viral vector expression, the specific neuronal population targeted, and the desired experimental outcomes. Our choice of 3.5 mg/kg was thus based on a combination of literature review and our laboratory's empirical data, aiming to achieve effective chemogenetic inhibition while minimizing potential off-target effects.

4. *The authors states that the inhibition of the SC-LPMR pathway has presynaptic effects only, but it would be important to investigate whether inhibition of postsynaptic LPMR neurons produces similar effects.*

➡ Thank you for your insightful comment regarding the investigation of postsynaptic effects in the inhibition of the SC-LPMR pathway. We agree that this aspect is crucial for a comprehensive understanding of the pathway's role in fear responses.

In our manuscript, we presented data (specifically in Fig. 3i and 3j) that directly addresses this concern. The chemogenetic inhibition of LPMR neurons elicited a similar increase in defensive responses to height threats as observed with the inhibition of SC neurons. Furthermore, our additional experiments involving the localized application of CNO in the LPMR to specifically inhibit the presynaptic terminals of the SC-LPMR projection also resulted in an increased fear of heights (Fig. 3m, n). This parallel in behavioral outcomes between presynaptic and postsynaptic interventions indicates that inhibition at both levels of the SC-LPMR pathway contributes to the modulation of fear responses to height. These new findings and further explanation have been incorporated in the revised manuscript (P. 7, lines 19-26).

5. *Considering that PV-positive neurons in SC can project to LPTN, it would be informative to compare the SC neurons projecting to LPMR and LPTN. Are there distinct subpopulations of SC neurons projecting to LPMR? How do these different populations of SC neurons become activated in response to looming or fear of heights*

➡ Thank you for prompting a detailed discussion on the relationship between SC neurons that project to the LPMR and those involved in mediating the looming response.

In response to your suggestion, we conducted a further analysis on the neuronal identity of SC neurons projecting to the LPMR. These neurons are predominantly situated in the medial region of the SC, as shown in Fig. 3m, with around 60% being Parvalbumin-positive (PV-positive) (Supplementary Fig. 6c). Additionally, our data demonstrated a significant increase in c-Fos-positive cells within the superficial layers of the medial SC following height exposure (Fig. 3a-d). Considering that LPMR is a subregion of LPTN and that the PV-positive neurons responsive to looming stimuli also project from the superficial layer of medial SC to LPTN, our findings suggest a potential overlap in the SC neuronal populations projecting to these interconnected areas. This implies that while there may be a shared pool of neurons in the medial SC area that respond to both looming and height stimuli and both project to the LPTN, the specific neuronal ensembles within these pathways may be selectively activated by different types of stimuli. The specific nature of the stimulus seems to play a

crucial role in triggering subsets of neurons, resulting in divergent behavioral outcomes. This selective activation underscores the complexity of neural responses to environmental stimuli and their impact on behavior. Further investigation is needed to validate this intriguing possibility and to unravel the detailed mechanisms behind the differential activation of these overlapping neuronal pathways. Further discussion on this intriguing topic has been incorporated in the revised manuscript (P. 13, lines 17-28).

6. It would be interesting to investigate whether direct activation of the SC-LPMR pathway can reduce fear. Additionally, can activation of the vLGN-PAG pathway induce trembling under conditions of height that do not induce fear?

➡ Thank you for your interest in the potential effects of directly activating the SC-LPMR and vLGN-PAG pathways. Your question aligns with a theme we previously discussed in response to a similar query from reviewer #1.

Activation of the SC-LPMR pathway: As mentioned in our response to reviewer #1 comment #3, we have begun investigating the effects of excitatory modulation in these pathways. Our preliminary experiments with chemogenetic activation of SC neurons suggest it might reduce fear of heights. However, these initial findings indicated responses that might not reflect normal physiological conditions, such as an exaggerated "flee" behavior in response to SC excitation (Video 1_for_Reviewers only). Therefore, these results were not included in the current manuscript. We are planning further studies with adjusted dosages and conditions with more specific activation of the SC-LPMR pathway to better understand the physiological relevance of this pathway's activation in fear of heights.

Activation of the vLGN-PAG pathway: Thank you for your suggestion to investigate whether activation of the vLGN-PAG pathway induces trembling in non-fear-inducing height conditions. We acknowledge the value of exploring this question for a deeper understanding of the pathway's role in height perception and response. In our preliminary experiments aimed at activating the vLGN, we observed intense behavioral responses in mice, resembling seizure-like behavior (Video 2_for_Reviewers only). This outcome suggests that the level of stimulation may have exceeded physiological norms, potentially skewing the interpretation of the pathway's role in typical fear responses. Therefore, we decided not to include these results in the current manuscript and plan to refine our experimental method for future studies.

In summary, our preliminary findings on circuit excitation, while intriguing, underscore the need for further investigation. We aim to conduct more refined studies to elucidate the role of these pathways in fear modulation, ensuring a focus on physiological relevance.

7. The quality of the fluorescence images of the brain slices, especially the c-Fos images induced by height, could be improved. Furthermore, inconsistencies in the scale and size of the brain slice images within the same brain area should be addressed. The scale bars in some of the brain slice images also appear to be incorrect.

➡ Thank you for your constructive feedback on the quality of the fluorescence images, particularly the c-Fos images, and your observation regarding inconsistencies in the scale and size of brain slice images. We appreciate your attention to these details. In response to your suggestion, we have carefully reviewed all the fluorescence images in our manuscript, especially the c-Fos images induced by height exposure. To enhance clarity and detail, we have reprocessed these images using optimized settings for zoom in scale, brightness, contrast, and resolution. This ensures that the features of interest are more visible and distinct. We have standardized the magnification and scale across all images. We have also thoroughly reviewed and corrected any inaccuracies.

8. Since the fear of heights in mice is more pronounced on the 2nd or 7th day, it would be interesting to investigate whether modulation of the two circuits during this period could block this fear. Are there any plasticity changes in the relevant brain areas?

➡ Thank you for your suggestion to investigate the circuit mechanism of the experience-dependent enhancement of fear of heights. We agree that exploring this aspect could provide valuable insights into the plasticity changes in the brain areas associated with the fear of heights and could significantly enhance our understanding of the adaptive and maladaptive responses to height-induced fear. However, due to time constraints and the extensive nature of the studies required to adequately address this question, we are unable to include such experiments in the current project. We do recognize the importance of this line of inquiry and believe it merits a dedicated study. Therefore, we are considering this as a potential focus for future research. In the revised manuscript, we have discussed this as an interesting avenue for future research and acknowledged the need for in-depth studies to unravel these dynamic aspects of fear modulation (P. 11, line 24 to P. 12, line 4).

9. Considering that vLGN also projects to SC (Fratzl et al., 2021), how does this relate to its connection with the SC-LP and LGN-SC pathway? How does this interplay contribute to the fear of heights in mice under physiological conditions?

➡ Thank you for highlighting the intricate interconnections between vLGN, SC, and

their associated pathways in the context of fear of heights. Your query delves into a fundamental aspect of the neural mechanisms underlying this behavior.

As outlined by Fratzl et al. (2021), the vLGN has projections to the SC, which are involved in modulating the animal's response to looming visual threats. Given that exposure to heights represents a similar form of visual threat, it is plausible that the vLGN-SC pathway might also play a role in the processing and perception of height, subsequently affecting the fear response. Our study specifically shows that activation of the vLGN mediates the fear of heights via its downstream target the PAG. In contrast, the SC appears to suppress the defensive response to height threats through the LPMR. Since the predominant output neurons of the vLGN are GABAergic, the vLGN-SC projection may provide feedforward inhibition to the SC, facilitating the expression of a defensive response to height threats. However, the precise contributions and interplay of these pathways in the context of fear of height remain topics for future research. This intriguing topic has been incorporated into the discussion section of the revised manuscript (P. 15, line 24 to P. 16, line 2).

Reviewer #3 (Remarks to the Author):

The authors investigate aversive behaviour to height stimuli in mice in order to obtain clues as to the neurophysiological background of the height vertigo-fear of height phenomenon.

The article is quite informative and there is definitely a lot of work behind it. The authors first phenotype the mouse-specific response to a height stimulus at the behavioural level. Then they quite systematically manipulate environmental stimulus conditions on the one hand and stimulus perception conditions on the other. In particular, they work out the central role of visual stimuli. In the next step, they identify components along the various neural visual processing networks that respond specifically to the height stimulus. In a final step, they selectively switch off the function of these components chemically or surgically and, with regard to potential changes at the behavioural level, are able to elaborate two visual circuits that very specifically link visual height stimuli and fear response. The writing style of the manuscript is very technical, but this is quite common for articles published by Nature Communications.

Major queries:

(1) In several places, the terms concerning physiological and non-physiological responses to altitude exposure are not used correctly (Introduction, Discussion). In

addition to a physiological response to altitude exposure affecting 100% of the population, there is a visual height intolerance that occurs in about 30% of the population, with a continuum to acrophobia affecting about 6% of the population (Brandt and Huppert, Curr. Opin., Neurol., 2014)

➔ We appreciate your attention to the accuracy of the terminology used in our manuscript, particularly in differentiating physiological responses from visual height intolerance and acrophobia. In response to your guidance, we have revised the relevant sections to accurately reflect these distinctions (P. 3, lines 15-21). We now clearly delineate the physiological response to altitude, which is universal, from visual height intolerance and acrophobia, highlighting the prevalence of each condition as reported in the literature.

(2) Fear of heights is described as a motion sickness ('Section 'vestibular input is dispensable...'; 2nd paragraph Discussion). This is not correct.

➔ We appreciate your attention to this detail and agree that this characterization is not accurate. Our current findings also suggest that they are fundamentally different. In light of your feedback, we have revised the relevant section of our manuscript to correctly describe fear of heights (P. 5, line 25 to P. 6, line 26).

(3) Generally speaking, the presentation of the results in the various subsections is very detailed, sometimes difficult to read. In this respect, a summary table on magnetic resonance imaging changes would be useful. In the Discussion section at the end, the magnetic resonance imaging changes are again presented in great detail. A less detailed discussion style would make more sense here.

➔ Upon careful examination, we consider that this comment would be a typographical error and is not relevant to our study.

(4) It is not entirely convincing that the quantified behavioural changes at altitude should be specific to fear of heights. For that, they would have to be contrasted with behavioural reactions to other fear stimuli, which the authors do not do. This aspect also fundamentally affects the other network analyses. There, too, the current methodological approach cannot convincingly show that the response is specific to a height stimulus and does not occur with other fear-inducing stimuli. Moreover, the description of how the individual behavioural parameters were quantified is only very cursory; as a reader, I would like to have more background information. I also wonder why autonomic responses (e.g. pulse increase) were not also measured; in earlier studies cited by the authors and in which they were also partly involved, ECG measurements were carried out in mice at altitude; this seems to be an established

and common methodology.

➡ Thank you for your insightful and constructive comments. We understand your concern regarding the specificity of the quantified behavioral changes to the fear of heights, and the necessity to differentiate these from reactions to other fear stimuli.

Comparative analysis with classical fear conditioning: Following your suggestion, we have incorporated a comparative analysis in our revised manuscript, contrasting mouse behaviors on an OHP with those elicited during classical fear conditioning. This analysis elucidates the unique trembling behavior observed in mice exposed to heights, distinctly different from the freezing behavior noted in fear conditioning (Supplementary Movie 2) and responses to looming threats (P. 3, lines 15-21).

In addition, further experiments involving local CNO infusion into the l/vIPAG to inhibit the presynaptic terminal of the vLGN-PAG pathway showed a significant reduction in the fear of heights, without altering responses in classical fear conditioning tests (P. 10, lines 18-22; Fig. 6o-q). This finding lends further support to the specific role of the vLGN-PAG pathway in height-induced fear, as opposed to a generalized role in aversive stimuli responses.

Expanded methods section: We also acknowledge the need for a more comprehensive description of our methods for quantifying behavioral parameters. Therefore, the Methods section has been expanded to include detailed descriptions of how we quantified various behaviors including locomotion, trembling, rearing, grooming, etc.

Inclusion of autonomic response measurements: Regarding the measurement of autonomic responses, we appreciate your suggestion of incorporating ECG measurements, which are indeed a valuable addition to our analysis. We have included data from EEG-based heart rate measurements in our revised manuscript to provide a more comprehensive view of the fear of heights (P. 3, line 32 to P. 4, line 1; P. 22, line 24 to P. 23, line 2; Supplementary Fig. 1).

(5) *The authors report the height-dependent saturation of fear of heights, similar to what Brandt described in the 1980s. In humans, virtual simulation was used to test the dependency of neurophysiological responses to different heights, where saturation was found (Wuehr et al., J. Neurol., 2019). However, this aspect is not taken up and discussed further by the authors. It would be interesting to see how this phenomenon can be explained in the context of the new findings and the authors' clear rejection of the sensory mismatch theory.*

➡ We are grateful for your valuable suggestion to include a discussion on the height-dependent saturation of fear of heights in our manuscript. In response, we have carefully expanded the revised manuscript to include a section to discuss the potential

neurophysiological mechanisms underlying this saturation effect, placing our findings in a broader context (P. 11, line 24 to P. 12, line 4).

We found that the behavioral response of mice to height exposure reaches saturation at approximately 20 cm. This mirrors the saturation phenomenon in human fear of heights reported in earlier studies, underscoring the physiological relevance of our experimental paradigm. It also provides a platform to understand how emotional responses to external stimuli, such as heights, may reach a saturation point. We believe that this saturation can occur at multiple levels and may be interpreted in a variety of ways.

Macroscopic perspective: We propose that the response to fear-inducing stimuli, such as heights, typically exhibits a non-linear pattern, with upper limit. This saturation in the fear response may serve as a protective mechanism, preventing excessive fear and allowing for adaptive responses crucial for survival.

Sensory input perspective: The sensory mismatch hypothesis, as eloquently proposed by Brandt and colleagues, offers an insightful explanation for the threshold and saturation phenomena in fear of heights, attributing it to the mismatch between visual and vestibular inputs. In contrast, our findings suggest an alternative mechanism. We hypothesize that visual information, particularly from the lower visual field, is pivotal in eliciting the fear of heights. As the height increases, the visual stimulus intensifies the fear response, but beyond a certain point, this sensory input may saturate, thereby limiting the extent of fear experienced.

Neuronal level perspective: At the cellular level, the non-linear membrane properties of neurons are likely to influence the input-output response in height perception, with a ceiling in their firing rate. Once this neuronal activity reaches its upper limit, the behavioral response to height threat may also reach a point of saturation.

Circuitry level perspective: Additionally, our study highlights the coexistence of both fear-enhancing and fear-reducing neural circuits, exemplified by the vLGN-PAG and SC-LP pathways. The interplay of these opposing neural regulations may further contribute to the saturation of emotional responses, thus preventing excessive fear.

In summary, our revised manuscript discussed the potential mechanisms behind the saturation phenomenon in the fear of heights, suggesting a multifaceted interplay of neural responses that contribute to this behavior.

(6) The authors examine the effect of repeated altitude exposure on the behavioural level and report an increased fear response to repeated altitude stimulation. This contradicts data in the literature, where a rather constant reaction could be found, and other reports from behavioural therapy. The authors should definitely discuss this aspect more, because it calls into question whether the animal model is really a

suitable model for human behavioural response.

➡ Thank you for bringing up the perceived contradiction between our findings and the existing literature, specifically referencing the study by Wang et al. (2021) and the principles of exposure therapy for acrophobia. We believe that the methodological differences between our study and others may contribute significantly to the apparent contradictions.

Comparison with Wang et al. (2021): In their study, Wang and colleagues focused on neuronal activity in the BLA in response to height exposure. While they did not observe significant changes in neuronal response with repeated exposure, it is crucial to note that their study did not investigate behavioral outcomes. As such, the question of whether behavioral adaptation or sensitization to height exposure occurs was not explored in their work. Our findings contribute to this discussion by showing that, although the BLA exhibits increased neuronal activity during height exposure, it is not the primary region triggering the fear of heights. We propose that the plastic alteration of neuronal activity in key circuits, such as the vLGN-PAG pathway, upon repeated exposure, is a vital area for future research.

Relation to exposure therapy in acrophobia: Concerning the practice of exposure therapy for acrophobia, it is indeed essential for the therapy to be conducted in a safe and controlled environment to prevent exacerbating the fear. Our experimental setup, which involved exposing mice to a threatening height repeatedly, differs from this therapeutic context. In a situation where the perceived threat is high, an increased aversive response is a reasonable outcome, which does not contradict the principles of exposure therapy. Our finding of the cross-session facilitation highlights the need for a careful design of the exposure protocol in exposure therapy to avoid reinforcing the fear through repetitive height exposure. Additional discussion on this point has been included in the revised manuscript (P. 11, line 18-23).

(7) The exposure to heights in the dark is difficult to understand. Overall, little detail is given, e.g. whether the mice were naïve or were able to explore the platform in the light beforehand; and if they were naïve, whether many mice then simply walked over the edge of the platform.

➡ Thank you for your valuable feedback highlighting the need for more detailed information regarding the exposure to heights in dark conditions. To address this, we have expanded the Methods and Results sections in the revised manuscript to provide a comprehensive explanation of this experimental setup.

Specifically, we have further clarified that the mice used in these experiments were naïve to the setup. They had no prior exposure to the high platform in a lit environment before being tested in the dark (P. 4, lines 26-27).

Additionally, we observed that naïve mice did not walk over the edge of the platform when exposed to heights in the dark. This indicates that without visual cues, the mice exhibited anxiety upon reaching the edge of the platform. This indicates that edge-associated anxiety includes both a fear of heights which depends on visual input and a vision-independent element likely linked to posture instability at the edge, potentially involving vestibular and proprioceptive inputs. It is important to note that this vision-independent anxiety should not be considered as fear of heights but may intensify the fear and anxiety at heights. We have discussed these observations and their implications in understanding the fear of heights under different sensory conditions on P.12, lines 24-30.

(8) Overall, the role of the individual behavioural tests (open field, elevated open maze, etc.) and the implications drawn from the test results are not entirely clear. The information given in the manuscript about this is very sparse. For example, why is the open field test an indicator of general anxiety levels? At first glance, this does not seem entirely plausible.

➡ Thank you for highlighting the need for clarity regarding the behavioral tests used in our study. To address this, we have expanded the Methods and Results sections to explain the rationale and significance of each test more thoroughly.

Regarding the open field test and elevated O-maze as indicators of general anxiety levels, we have included additional references and a more detailed explanation to substantiate its relevance and validity (P. 6, lines 1-2; P. 17, lines 8-9 and lines 17-21).

The open field test is a well-established method in behavioral neuroscience for assessing anxiety-like behaviors in rodents, primarily based on their natural aversion to open and brightly lit areas. In this test, the level of anxiety is inferred from the animal's propensity to explore the central area of a square open field arena, with lesser exploration indicating higher anxiety.

The elevated O-maze test is another prominent behavioral paradigm for assessing anxiety-like behavior in rodents. This test involves a circular track, elevated above the ground, featuring alternating open and enclosed sections. The open sections lack walls, while the enclosed sections have high walls. Anxious animals typically exhibit a preference for the enclosed, safer sections over the exposed, open areas. Researchers measure the time spent in and the number of entries into each section to evaluate anxiety levels in the test subjects.

(9) Some abbreviations in the text are not introduced, for example CeA.

➡ Thank you for your careful review and for highlighting the oversight regarding the

use of certain abbreviations without proper introduction. In response to your feedback, we have thoroughly reviewed the manuscript and ensured that all abbreviations, including CeA (Central Amygdala), are properly introduced and defined at their first occurrence.

REVIEWERS' COMMENTS

Reviewer #1 (Remarks to the Author):

All my concerns have been addressed. Thank you very much!

Reviewer #2 (Remarks to the Author):

The authors have addressed my concerns. This reviewer has no further comments.

Reviewer #3 (Remarks to the Author):

The authors have worked on the points raised in detail and changed the corresponding passages in the manuscript so that the more differentiated presentation of the various reactions to exposure is now easier to understand.

The term 'physiological fear of heights, also known as height vertigo' in lines 3 and 4 of the introduction should be changed to 'physiological reaction to exposure of heights is considered...', for example.

In the Discussion, page 11, line 7, the term 'height vertigo' is used; this is an unspecific term, should be replaced according to the definitions of reaction to height that the authors have now presented in the Introduction

Apart from that, I have no further objections

REVIEWERS' COMMENTS

Reviewer #1 (Remarks to the Author):

All my concerns have been addressed. Thank you very much!

➡ Thanks for your constructive comments that make our data more promising.

Reviewer #2 (Remarks to the Author):

The authors have addressed my concerns. This reviewer has no further comments.

➡ Thanks for your constructive comments that make our data more promising.

Reviewer #3 (Remarks to the Author):

The authors have worked on the points raised in detail and changed the corresponding passages in the manuscript so that the more differentiated presentation of the various reactions to exposure is now easier to understand.

The term 'physiological fear of heights, also known as height vertigo' in lines 3 and 4 of the introduction should be changed to 'physiological reaction to exposure of heights is considered...', for example.

In the Discussion, page 11, line 7, the term 'height vertigo' is used; this is an unspecific term, should be replaced according to the definitions of reaction to height that the authors have now presented in the Introduction

Apart from that, I have no further objections

➡ We have corrected our expression according to the reviewer's instructions:

1. page 2, lines 3 and 5: The physiological reaction to height exposure is considered an innate behavior that serves as a protective mechanism against falling-related injuries².
2. page 11, lines 3-5: Thus, mice serve as an excellent animal model for dissecting the neural mechanisms of height fear and for developing translational approaches to address acrophobia.